# Prototype-guided Bilateral Alignment Multimodal Federated Learning

**Tianchi Liao**[1 2]  **Lele Fu**[2]  **Sheng Huang**[2]  **Qing Hu**[2]  **Hong-Ning Dai**[1]  **Chuan Chen**[2]

## Abstract

Multimodal federated learning (MFL) has emerged as a pivotal paradigm for leveraging distributed data to enhance model performance. However, existing methods predominantly rely on idealized assumptions of model homogeneity and balanced modality distributions, rendering them ill-suited for practical scenarios characterized by heterogeneous client architectures and severe modality imbalance. To address these challenges, we propose a **M**ultimodal **Fed**erated learning **P**rototype-guided **B**ilateral **A**lignment (MFedPBA) framework. MFedPBA facilitates robust knowledge synergy through a dual alignment mechanism: (i) at the feature level, it aligns heterogeneous feature spaces via a projection encoder optimized by contrastive learning and the Gromov-Wasserstein distance; (ii) at the decision level, it employs an entropy-weighted aggregation of naturally aligned logit prototypes. This novel design achieves robust MFL by jointly tackling heterogeneous feature spaces and collectively aggregating decisions. Extensive experiments demonstrate that our method significantly outperforms state-of-the-art baselines under conditions of model heterogeneity and modality imbalance.

## 1. Introduction

With the rapid proliferation of multi-sensor Internet of Things devices and the growing demand for data privacy, personalized federated learning (FL) has emerged as an important paradigm for training customized local models on decentralized clients (Collins et al., 2021; Meng et al., 2024; Kou et al., 2026; Ma et al.). Recently, the integration of heterogeneous sensory data has further advanced multimodal federated learning (MFL), which aims to ex-

ploit complementary information across modalities such as vision, audio, and text to improve model robustness and personalization (Qi et al., 2023; Che et al., 2023; Huang et al., 2024; 2026). However, the majority of existing multimodal federated learning frameworks are predicated on the idealized assumption of model homogeneity, mandating that all participating clients deploy an identical neural network architecture (Hu et al., 2024b; Liao et al., 2024; Cao et al., 2026). In practice, as illustrated in Figure 1(a), this assumption rarely holds true in real-world deployments. Given that practical clients typically exhibit profound heterogeneity in both hardware resource constraints (e.g., computational and storage capacities) and multimodal data distributions (such as missing modalities or data skewness), a single global model struggles to simultaneously accommodate the diverse conditions of all nodes. Consequently, designing and deploying personalized, heterogeneous model architectures tailored to different clients has emerged as an inevitable trend (Chen & Zhang, 2024; Rahman & Nguyen, 2024).

To address model heterogeneity, prior work has explored prototype-based federated learning, where class embeddings are exchanged as communication units (Tan et al., 2022a; Huang et al., 2023; Li et al., 2025). Nevertheless, such approaches encounter severe bottlenecks in multimodal settings. Since heterogeneous encoders map multimodal data into disparate embedding spaces, the direct aggregation of prototypes on the server results in significant representation misalignment (Fu et al., 2025; Zhang et al., 2025c). As illustrated in Figure 1(b), when clients employ different encoders, prototypes corresponding to the same class may reside in incompatible feature spaces. Consequently, forced aggregation causes the global prototype to be close to class decision boundaries. Instead of yielding performance gains, such forced aggregation can lead to significant knowledge contamination. This issue often causes the global model to degrade to the point where its performance falls below that of a baseline trained purely on local data (Li et al., 2024; Chen et al., 2025; Xiao et al., 2026; Hu et al., 2024a). This raises a critical research challenge: *How can we overcome feature space incompatibility to facilitate effective knowledge transfer when both client neural architectures and input modality configurations are inconsistent?*

Furthermore, existing MFL research typically operates under the assumption of balanced modality distributions.

---

[1]Department of Computer Science, Hong Kong Baptist University, Hong Kong SAR, China [2]School of Computer Science and Engineering, Sun Yat-Sen University, Guangzhou, China. Correspondence to: Chuan Chen <chenchuan@mail.sysu.edu.cn>.

*Proceedings of the $43^{rd}$ International Conference on Machine Learning*, Seoul, South Korea. PMLR 306, 2026. Copyright 2026 by the author(s).

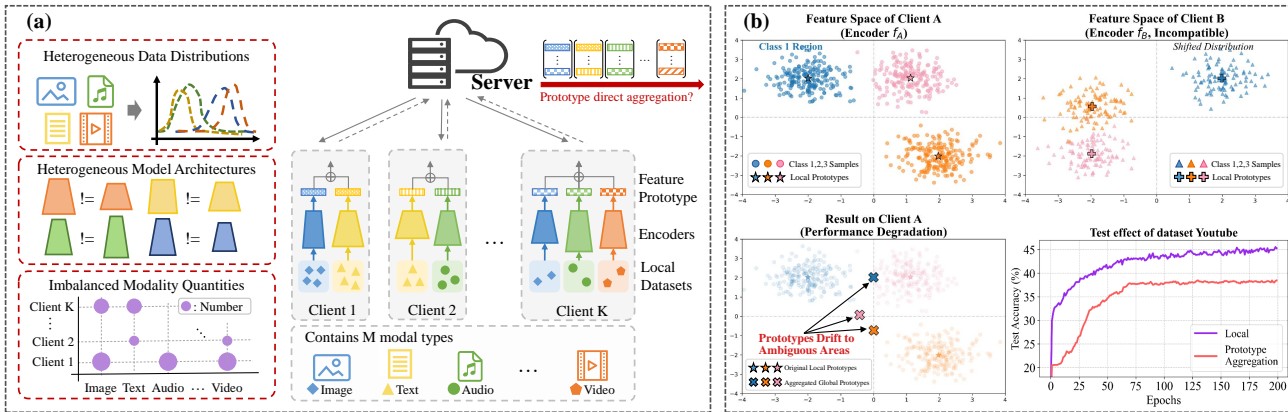

*Figure 1.* (a) Illustration of the multimodal federated learning setting with complex architectural and data inconsistencies. (b) Visualization of feature space incompatibility across heterogeneous encoders, where direct prototype aggregation induces drift towards ambiguous regions, resulting in performance inferior to local training.

While some methods consider scenarios with missing modalities, they are often restricted to bimodal settings or assume that all clients share an identical set of modalities (Che et al., 2024; Wang et al., 2024; Pan et al., 2025; Yu et al., 2025). Such assumptions fail to capture the prevalent imbalance in modality availability across decentralized clients. In real-world applications, systems commonly involve multiple modalities, and certain clients may entirely lack data for specific modalities. This imbalance substantially increases the difficulty of collaborative learning by introducing a higher degree of information heterogeneity (Qi et al., 2025; He et al., 2025). This leads to another key challenge: *How can we design an alignment mechanism that ensures precise collaboration in complex scenarios characterized by unbalanced data distributions and modality quantities exceeding the bimodal limit?*

To address these challenges, we propose a multimodal federated learning prototype-guided bilateral alignment framework (MFedPBA). The framework implements a bilateral alignment mechanism on the server that integrates knowledge from heterogeneous clients through modality-specific feature prototypes and logit prototypes. ❶ At the feature level, the server constructs an autoencoding alignment framework. It leverages contrastive learning to enhance intra-modal semantic consistency, while employing the Gromov–Wasserstein (GW) distance to align the geometric structures between heterogeneous feature spaces and the global space, enabling effective aggregation within a unified representation space. ❷ At the decision level, unlike feature prototypes that may reside in incompatible spaces, logits naturally reside in a shared semantic space aligned across heterogeneous models, as each dimension directly corresponds to a specific class. Leveraging this property, we employ an entropy-based weighting strategy during aggregation. This mechanism quantifies the reliability of local predictions to construct highly discriminative global logit

prototypes. By synergizing feature-level and decision-level insights, our framework facilitates robust and efficient cross-client knowledge transfer while strictly preserving the architectural heterogeneity of local models.

The primary contributions of this work are summarized as follows:

- **New perspective:** We address a realistic MFL setting characterized by model heterogeneity and modality imbalance, transcending conventional homogeneous assumptions to enable robust knowledge transfer.

- **Effective algorithm:** We propose MFedPBA, a bilateral alignment framework that achieves dual-level synergy: feature alignment via GW distance and contrastive learning, and decision alignment via entropy-based logit aggregation. We further provide theoretical convergence guaranties in non-convex settings.

- **Superior performance:** Extensive experiments on multiple multimodal benchmarks demonstrate that MFedPBA consistently outperforms state-of-the-art baselines under diverse modality distribution settings.

## 2. Related Work

### 2.1. Multimodal Federated Learning

Multimodal federated learning (Feng et al., 2023; Ouyang et al., 2023; Pan et al., 2024) has emerged as a pivotal paradigm for addressing privacy concerns in mobile sensing systems, finding broad application in real-world scenarios (Zhao et al., 2022; Zheng et al., 2023; Thrasher et al., 2025). Existing MFL literature predominantly addresses two settings: (1) *unimodal clients*, where each client possesses only a single modality (Zhang et al., 2025b; Deng et al., 2025), and (2) *missing modalities*, where clients hold in-

complete subsets of views (Che et al., 2024; Liu et al., 2025). While these approaches aim to aggregate effective global models, real-world deployments often necessitate distinct feature extractors tailored to specific modalities (Chen et al., 2024). To handle such heterogeneity, recent work like FedM-Bridge (Chen & Zhang, 2024) proposes a topology-aware hypernetwork. However, this method incurs prohibitive communication overhead, which scales poorly with increasing model complexity and modality counts. Consequently, existing methods fail to adequately address the compounding challenges of inter-node modal imbalance and model heterogeneity.

However, such methods incur prohibitive communication overhead and suffer from severe scalability bottlenecks as model complexity and the number of modalities increase. In contrast, our work proposes a communication-efficient heterogeneous federated learning framework via multimodal joint alignment.

## 2.2. Federated Prototype Learning

Federated prototype learning has been extensively investigated across diverse tasks, wherein a prototype is defined as the mean feature vector of instances belonging to a specific class (Huang et al., 2022; Qi et al., 2025; Liao et al., 2025). Within the federated learning literature, prototypes serve as a vital mechanism to abstract knowledge while strictly preserving data privacy (Tan et al., 2022b; Zhao et al., 2024). Due to their robust representational capacity, prototypes are widely adopted to enforce local regularization and enhance communication efficiency (Huang et al., 2023; Yi et al., 2023). To mitigate the feature drift issue in FL, (Fu et al., 2025) proposed a federated domain-independent prototype learning approach, which achieves representation and parameter space alignment under feature shifts. Furthermore, FedPall (Zhang et al., 2025c) leverages prototype-based adversarial learning to unify the feature space and employs collaborative learning to reinforce class-specific information within the features. Conventionally, local prototypes are simply aggregated into a global prototype via weighted averaging on the server, yielding sub-optimal global knowledge that negatively impacts client performance. To address this limitation, FedTGP (Zhang et al., 2024) employs adaptive margin-enhanced contrastive learning to optimize trainable global prototypes at the server level.

However, the feature misalignment arising from multimodal heterogeneity is significantly more severe than standard domain shifts. Consequently, naive aggregation fails to bridge the divergent feature spaces generated by heterogeneous local models. To address this challenge, we propose a learnable prototype framework on the server. By distilling knowledge from client-side prototypes, our approach effectively aligns the heterogeneous feature spaces.

## 3. Preliminary

### 3.1. General multimodal FL Framework

Consider a heterogeneous MFL problem. The system consists of a central server and $K$ clients. There are a total of $M$ modality types (e.g. image, video, text, and audio, etc.) and $C$ classes globally. The $k$-th client only possesses data from a subset of modalities $\mathcal{M}_k \subseteq \{1, \ldots, M\}$, and has a combinatorial input space $\mathcal{X}_{\mathcal{M}_k} = (\mathcal{X}_m | \forall m \in \mathcal{M}_k)$, where $\mathcal{X}_m$ is the subspace associated with the modality type $m$. The data distribution, quantity, and modality configuration of different clients are inconsistent. For client $k$ and modality $m \in \mathcal{M}_k$, its local data is denoted as $\mathcal{D}_k = \{(\boldsymbol{x}_{k,i}, y_{k,i})\}_{i=1}^{N_k}$, where $y_{k,i} \in \{1, \ldots, C\}$. Each sample's input consists of the modalities $\boldsymbol{x}_{k,i} = (\boldsymbol{x}_{k,i}^m)_{m \in \mathcal{M}_k}$ present as in $\mathcal{M}_k$, where $\boldsymbol{x}_{k,i}^m$ denotes the modality $m$ in $\boldsymbol{x}_{k,i}$.

Following previous conventions (Tan et al., 2022a), we split each client $k$'s model $\theta_k$ into a feature extractor $f_k$ parameterized by $\varphi_k$ and a classifier $h_k$ parameterized by $w_k$. Each client is equipped with $|\mathcal{M}_k|$ modality specific feature extractors, denoted as $f_k^m : \mathcal{X}_{\mathcal{M}_k} \to \mathbb{R}^{d_D}$. The client possesses a single classifier, denoted as $h_k : \mathbb{R}^{d_D} \to \mathbb{R}^{d_C}$. For inference or training, the client concatenates the feature vectors output by all modality specific feature extractors, and feeds this into the classifier $h_k$ to obtain the final prediction. The global objective of MFL is formulated as:

$$\min_\theta \frac{1}{K} \sum_{k=1}^K \frac{1}{N_k} \sum_{i=1}^{N_k} \mathcal{L}_k(\theta_k) + \Omega(\theta_1, \theta_2, \cdots, \theta_K), \quad (1)$$

which aims to jointly optimize the local objectives of all clients $\min_{\theta_k} \mathcal{L}_k(\theta_k) = \mathbb{E}_{(\boldsymbol{x},y) \sim \mathcal{D}_k} l\left(y, f_k(\boldsymbol{x}; \theta_k)\right)$, where $l(\cdot, \cdot)$ is the loss function, and Eq. (1) utilizes a central server to encourage privacy-preserving knowledge sharing schemes $\Omega(\cdot)$ among clients in order to improve the local model performance of each client.

### 3.2. Prototype-based Federated Learning

In contrast to conventional FL, which relies on aggregating model parameters, prototype-based FL achieves knowledge sharing by exchanging class prototypes between clients and the server, thereby offering a viable solution for scenarios involving model heterogeneity.

**Feature-based prototypes.** During the local training phase, each client $k$ computes its local prototype for each class $c$ of each modality, using the following method:

$$E_{k,c}^m = \frac{1}{|\mathcal{D}_{k,c}^m|} \sum_{(\boldsymbol{x},y) \in \mathcal{D}_{k,c}^m} f_k^m(\boldsymbol{x}), \quad (2)$$

where $\mathcal{D}_{k,c}^m$ represents a subset of $\mathcal{D}_k^m$ for the $m$-th modality of the local dataset, containing all data samples belonging

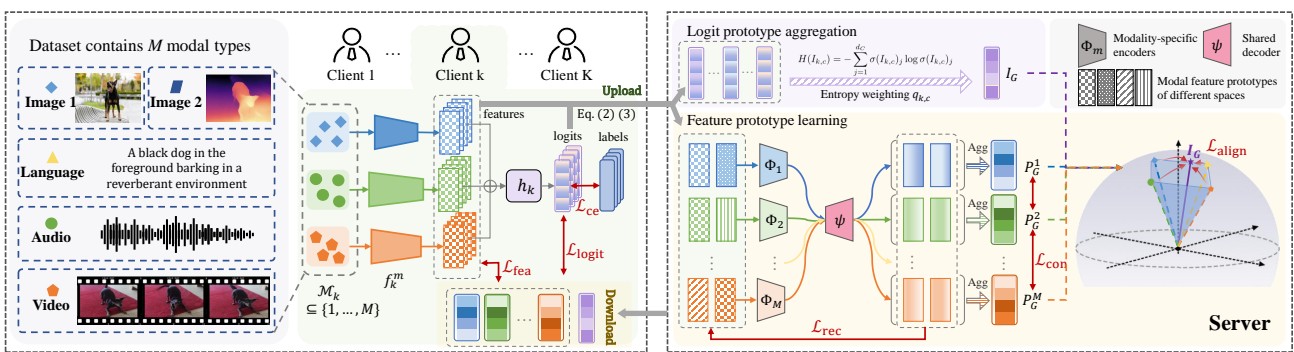

*Figure 2.* The framework of the MFedPBA.

to class $c$. Previous studies commonly upload local feature prototypes to the server and aggregate them via weighted averaging. However, in MFL, such direct aggregation is challenging due to heterogeneous feature extractors that project data into incompatible embedding spaces.

**Logit-based prototypes.** In contrast to feature prototypes, which suffer from inherent space incompatibility across heterogeneous models, logit prototypes reside in a naturally aligned shared semantic space. Each dimension of a logit prototype directly corresponds to a specific class, thereby providing intrinsic semantic alignment across heterogeneous models. This property effectively avoids the collaborative degradation caused by inconsistent embedding spaces and enables more robust knowledge aggregation in heterogeneous federated learning scenarios. Therefore, the prototype based on logits can be defined as the average of the logit vectors of all nodes under category $c$:

$$I_{k,c} = \frac{1}{|\mathcal{D}_{k,c}^m|} \sum_{(\boldsymbol{x},y)\in\mathcal{D}_{k,c}^m} h_k(f_k^m(\boldsymbol{x})). \quad (3)$$

Considering that logits naturally occupy a common semantic space irrespective of the underlying model architectures, the direct alignment of output-level logit prototypes emerges as a more rational approach compared to utilizing feature-level prototypes.

### 3.3. Gromov Wasserstein Distance

The Gromov-Wasserstein (GW) distance provides a rigorous framework for quantifying the discrepancy between two metric measure spaces (Mémoli, 2011). The GW distance is particularly well-suited for heterogeneous settings where a direct correspondence between feature dimensions is unavailable, which facilitates alignment by comparing the intrinsic geometric or relational structures of the data (Saha et al., 2025). Specifically, the GW framework evaluates the internal pairwise distance distributions within each space and seeks an optimal coupling (transport plan) that minimizes the relational distortion between these topologies.

Formally, let $(\mathcal{X}, C_X, \mathbf{p})$ and $(\mathcal{Y}, C_Y, \mathbf{q})$ be two metric measure spaces, where $C_X, C_Y$ denote the distance metrics of data $\mathcal{X}$ and $\mathcal{Y}$, respectively (Peyré et al., 2016). The $\mathbf{p}, \mathbf{q}$ represent the probability measures associated with each domain. The GW distance is defined as:

$$GW_p(C_X, C_Y, \mathbf{p}, \mathbf{q}) =$$

$$\left( \min_{T \in \Pi(\mathbf{p},\mathbf{q})} \sum_{i,j,k,l} L(C_X(i,k),\ C_Y(j,l))^p\, T_{ij} T_{kl} \right)^{1/p}, \quad (4)$$

where $\Pi(\mathbf{p},\mathbf{q}) = \left\{ T \in \mathbb{R}_+^{n\times m} \,\middle|\, T\mathbf{1}_m = \mathbf{p},\ T^\top \mathbf{1}_n = \mathbf{q} \right\}$ is a joint distribution of all couplings from $\mathbf{p}$ to $\mathbf{q}$, and $p$ is the order of distance (commonly $p$=2).

## 4. Methodology

The proposed MFedPBA aligns prototypes by collecting two types at the server. The MFedPBA framework is depicted in Figure 2, with the following details.

### 4.1. Global Prototype Modeling

Each client first performs local training and then collects its local prototypes, denoted as $\{E_{k,c}^m\}_{k\in K}$ and $\{I_{k,c}\}_{k\in K}$, according to Eq. (2) and (3), and uploads them to the server. Upon receiving the bilateral prototypes from all clients, the server constructs global prototypes at both the logit level and the feature level, respectively.

#### 4.1.1. Logit Prototype Aggregation
Since logits are the final outputs of classifiers, they naturally reside in a semantically aligned shared representation space across heterogeneous models. Therefore, no additional processing is applied to the logit prototypes, and they are directly aggregated. Considering that data distribution discrepancies across clients lead to varying levels of predictive uncertainty in their logit prototypes, an entropy weighted aggregation strategy is adopted. Clients with lower predictive uncertainty, corresponding to lower entropy, are assigned higher weights, resulting in a more reliable global

logit prototype. Accordingly, defining $\sigma(\cdot)$ as the softmax, the entropy of a logit prototype is defined as:

$$H(I_{k,c}) = -\sum_{j=1}^{d_C} \sigma(I_{k,c})_j \log \sigma(I_{k,c})_j. \tag{5}$$

The server weights the values based on the inverse of the entropy, obtaining a global logit prototype for class $c$:

$$\mathbf{I}_{G,c} = \sum_{k=1}^{K} q_{k,c} I_{k,c}, \quad q_{k,c} = \frac{(H(I_{k,c}) + \epsilon)^{-1}}{\sum_{k'}^{K} (H(I_{k',c}) + \epsilon)^{-1}}, \tag{6}$$

where $\epsilon$ is a very small positive constant that avoids numerical collapse.

### 4.1.2. Feature Prototype Learning

To effectively capture modality-specific discriminative characteristics and facilitate cross-modal collaboration, the server introduces an auto-encoding framework composed of modality-specific encoders $\Phi_m(\cdot) : \mathbb{R}^{d_D} \to \mathbb{R}^{d_D/2}$ and a shared decoder $\psi(\cdot) : \mathbb{R}^{d_D/2} \to \mathbb{R}^{d_D}$. This projects feature prototypes of varying modalities and dimensionalities into a unified and comparable semantic space, while enforcing multiple constraints to achieve representation alignment across both clients and modalities. Therefore, we have:

$$P_{k,c}^m = \psi(\Phi_m(E_{k,c}^m)). \tag{7}$$

The modality-specific encoders project features of the same modality from different clients into a unified representation space, while the shared decoder enforces a common semantic structure during reconstruction. This promotes cross-modal consistency and constraining latent representations from different modalities within a shared and information-rich semantic space. Consequently, global modality feature prototypes are derived through the weighted aggregation of these intra-modality aligned prototypes:

$$\mathbf{P}_{G,c}^m = \frac{1}{\sum_k \mathbb{I}(m \in \mathcal{M}_k)} \sum_{k,m \in \mathcal{M}_k} P_{k,c}^m, \tag{8}$$

here $\mathbb{I}$ denotes the indicator function.

To make the learned global modal prototypes more discriminative, the server updates the network parameters by minimizing the following joint loss function:

**Prototype Reconstruction Loss**: This loss ensures that the projection and reconstruction process preserves the essential information of the input prototypes:

$$\mathcal{L}_{\text{rec}} = \frac{1}{|K|} \sum_{k,m,c} \| P_{k,c}^m - E_{k,c}^m \|^2. \tag{9}$$

**Inter-modal Contrastive Loss**: This loss enhances semantic consistency across modalities within the same class by

pulling together projected features of different modalities for class $c$, while pushing apart features from different classes

$$\mathcal{L}_{\text{con}} = -\sum_c \sum_{m \neq m'} \log \frac{\exp\left(\text{sim}\left(\mathbf{P}_{G,c}^m, \mathbf{P}_{G,c}^{m'}\right)/\tau\right)}{\sum_{c'} \exp\left(\text{sim}\left(\mathbf{P}_{G,c}^m, \mathbf{P}_{G,c'}^{m'}\right)/\tau\right)}, \tag{10}$$

where $\text{sim}(u,v) = u\top v/(\|u\|\|v\|)$, and $\tau$ is the temperature coefficient.

**Intra-modal Distribution Alignment Loss**: This loss aligns the structural consistency of the same class across different modalities using the GW distance. This loss does not require strict dimensional alignment but preserves the proportional relationships of inter-class distances between the feature space and the logit space, thereby maintaining topological consistency.

We compute the GW distance between the modality-specific feature prototypes $\mathbf{P}_{G,c}^m$ with the global logit prototypes $\mathbf{I}_{G,c}$ space. Within each metric space (Mémoli, 2011), we characterize the intrinsic geometric topology using the pairwise squared Euclidean distance, defined as $C_{i,j} = \|x_i - x_j\|_2^2$. Consequently, we derive the cost matrix $C_{\mathbf{I}_G} \in \mathbb{R}^{d_C \times d_C}$ and $C_{\mathbf{P}_G^m} \in \mathbb{R}^{d_C \times d_C}$ to represent the inter-class relational structures within the feature and logit spaces, respectively. Therefore, this distance loss can be expressed as:

$$\mathcal{L}_{\text{align}} = \sum_m \left( GW_2^2(C_{\mathbf{I}_G}, C_{\mathbf{P}_G^m}, \mathbf{p}, \mathbf{q}) - \varepsilon H(T^m) \right)$$
$$= \sum_m \min_{T^m \in \Pi(\mathbf{p},\mathbf{q})} \sum_{i,j,k,l} \left( C_{\mathbf{I}_G}(i,k) - C_{\mathbf{P}_G^m}(j,l) \right)^2 T_{ij}^m T_{kl}^m$$
$$+ \sum_m \varepsilon \sum_{ij} T_{ij}^m \log T_{ij}^m. \tag{11}$$

To solve this problem, we introduce entropy regularization $H(T)$ to make the problem differentiable, where $\varepsilon$ weights this regularization, and then use the Sinkhorn algorithm (Cuturi, 2013; Séjourné et al., 2021) to solve the optimal transmission problem of this regularization.

Therefore, the server optimizes the global loss for $S$ rounds. The total loss is formulated as:

$$\mathcal{L}_{\text{server}} = \mathcal{L}_{\text{rec}} + \mathcal{L}_{\text{con}} + \mathcal{L}_{\text{align}}. \tag{12}$$

The refined client feature prototypes incorporate richer global semantics and modality-specific information. Therefore, the server constructs a global feature prototype for each modality $m$ by averaging the learned prototypes.

Finally, the server sends the global prototypes $\{\mathbf{P}_{G,c}^m\}_{m=1}^M$ and $\mathbf{I}_{G,c}$ to each client, transferring complementary cross-modal knowledge to compensate for the information deficiency in clients with missing modalities.

## 4.2. Client Local Update

Upon receiving the prototype set from the server, the objective of local training is to effectively distill knowledge from both the global feature prototypes and the global logit prototypes at the representation and decision levels, thereby maximally injecting global information into local representation learning. To this end, we propose a prototype-based supervised contrastive loss composed of two terms.

**Feature alignment loss** is introduced to mitigate feature space drift caused by model heterogeneity and biased local data distributions. Since the local feature extractor $f_k^m(\cdot)$ tends to overfit limited modal client data, we directly anchor local feature representations to the global feature prototype space, encouraging all clients to optimize toward a shared and consensus feature distribution. Specifically, we employ the mean squared error as the alignment metric, defined as

$$\mathcal{L}_{\text{fea}} = \frac{1}{|\mathcal{M}_k|} \sum_{m \in \mathcal{M}_k} \frac{1}{C} \sum_{c=1}^{C} \|\mathbf{P}_{G,c}^m - E_{k,c}^m\|_2^2. \quad (13)$$

**Logit alignment loss** performs knowledge distillation at the decision level. The global logit prototypes $\mathbf{I}_{G,c}$ encode rich information about inter-class relationships and global decision structures. By enforcing the local classifier outputs to match the global logit prototypes, robust and transferable decision knowledge is distilled into local models. Accordingly, we measure the distribution discrepancy using the Kullback–Leibler (KL) divergence:

$$\mathcal{L}_{\text{logit}} = \frac{1}{C} \sum_{c=1}^{C} \mathbb{D}_{\text{KL}}(\mathbf{I}_{G,c} \| I_{k,c}), \quad (14)$$

where $\mathbb{D}_{\text{KL}}(P \| Q) = \sum_i P(i) \ln(\frac{P(i)}{Q(i)})$. Finally, to preserve discriminative capability on local data domains, we incorporate the standard cross-entropy loss between the predicted logits and ground-truth labels:

$$\mathcal{L}_{\text{ce}} = \sum_{(\boldsymbol{x}_i, y_i) \in \mathcal{D}_k} -\mathbf{1}_{y_i} \log(h_k(f_k^m(x_i))), \quad (15)$$

where $\sigma(\cdot)$ is the softmax function. The total local loss is given as follows:

$$\mathcal{L}_{\text{client}} = \mathcal{L}_{\text{ce}} + \lambda_1 \mathcal{L}_{\text{fea}} + \lambda_2 \mathcal{L}_{\text{logit}}. \quad (16)$$

The overall MFL algorithm is shown in Algorithm 1.

## 5. Convergence Analysis

To analyze the convergence of MFedPBA, we define $t$ as the current communication round, $e \in \{0, 1, \cdots, E\}$ as the number of local iterations, where $E$ denotes the maximum number of local iterations. Thus, $(tE + e)$ represents the $e$-th iteration in the $(t+1)$-th communication round. We make

some assumptions see Appendix B.1. Based on the above assumptions, we have the following lemmas and theorems:

**Lemma 5.1.** *Based on Assumption B.1 and B.2, in the local iteration $e \in \{0, 1, ..., E\}$ of the $(t+1)$-th training round, the local model loss of any client is bounded by:*

$$\mathbb{E}\left[\mathcal{L}_{(t+1)E}\right] \leq$$
$$\mathcal{L}_{tE+0} - (\eta_c - \frac{L\eta_c^2}{2}) \sum_{e=0}^{E} \|\nabla \mathcal{L}_{tE+e}\|_2^2 + \frac{LE\eta_c^2}{2}\sigma^2. \quad (17)$$

**Lemma 5.2.** *After feature prototype learning and logit prototype aggregation are completed on the server, the loss function of any client can be constrained as follows:*

$$\mathbb{E}[\mathcal{L}_{(t+1)E+0}] \leq \mathcal{L}_{(t+1)E}$$
$$+ (M\lambda_1 + \lambda_2)L_r E\eta_c G_c + 2\lambda_1(\delta_s + MG_s). \quad (18)$$

Based on Lemma 5.1 and Lemma 5.2, we can further derive the following theorems.

**Theorem 5.3.** *The above assumptions, the expectation of the loss of an arbitrary client's local model before the start of a round of local iteration satisfies:*

$$\mathbb{E}[\mathcal{L}_{(t+1)E+0}] \leq \mathcal{L}_{tE+0} - (\eta_c - \frac{L\eta_c^2}{2}) \sum_{e=0}^{E} \|\nabla \mathcal{L}_{tE+e}\|_2^2$$
$$+ \frac{LE\eta_c^2}{2}\sigma^2 + \Gamma_d E\eta_c + \Gamma_s. \quad (19)$$

*where $\Gamma_d = (M\lambda_1 + \lambda_2)L_r G_c$ and $\Gamma_s = 2\lambda_1(\delta_s + MG_s)$ represent the drift constant and server bias constant.*

**Theorem 5.4.** *The above assumptions, for arbitrary client and any $\epsilon > 0$, if learning rate $\eta_c < \min\left\{\frac{2}{L}, \frac{2(\epsilon - \Gamma_d E)}{L(\epsilon + E\sigma^2)}\right\}$ and $\Gamma_s \to 0$, the following inequality holds:*

$$\frac{1}{T} \sum_{t=0}^{T-1} \sum_{e=0}^{E} \mathbb{E}\left[\|\mathcal{L}_{tE+e}\|_2^2\right] \leq \frac{2(\mathcal{L}_{t=1} - \mathcal{L}^*)}{T\eta_c(2 - L\eta_c)}$$
$$+ \frac{LE\eta_c^2\sigma^2 + 2(\Gamma_d E\eta_c + \Gamma_s)}{2\eta_c - L\eta_c^2}. \quad (20)$$

With this, it is evident that the local model of any client of MFedPBA converges at a non-convex convergence rate $\mathcal{O}\left(\frac{1}{T}\right)$. See Appendix B for a detailed proof.

## 6. Experiments

### 6.1. Experimental Setup

**Datasets and Model:** We used four benchmark datasets for experiments: Caltech101 [1], Reuters [2], NUS-WIDE [3], and

---

[1]https://data.caltech.edu/records/mzrjq-6wc02.
[2]https://www.kaggle.com/datasets/nltkdata/reuters.
[3]https://www.kaggle.com/datasets/xinleili/nuswide.

Youtube [4]. The data information is shown in Table 1. All datasets are divided into training and test sets in a ratio of 75%/25%.

*Table 1.* Dataset statistics. Acronyms for modality types: I (Image), L (Language), A (Audio), T (Time-series), X1 (Style-one X), X2 (Style-two X). The #S: Samples, #C: Classes, #M: Modalities.

| Datasets | Types | #S | #C | #M | Input modalities |
|---|---|---|---|---|---|
| Caltech101 | {I} | 9144 | 102 | 3 | {I1}, {I2}, and {I3} |
| Reuters | {L} | 18758 | 6 | 5 | {L1}, {L2}, {L3} {L4}, and {L5} |
| NUS-WIDE | {I, L} | 5000 | 10 | 4 | {I1}, {I2}, {I3}, and {L} |
| Youtube | {I, A, T} | 2000 | 10 | 6 | {I1}, {I2}, {T} {A1}, {A2}, and {A3} |

To model heterogeneous MFL scenarios, under the assumption that the number of clients $K$ equals the number of modalities $M$, we consider three client data distribution settings: "M2", where each client possesses two modalities; "M1+", where each client has at least one modality; and "M1", where each client contains exactly one modality. Using the YouTube dataset as an example, the corresponding data distributions are illustrated in Figure 3. We assign heterogeneous models to each modality of every client to simulate realistic model heterogeneity. Detailed descriptions of the local dataset partitions and the corresponding neural architecture configurations are provided in Appendix C.1 & C.2.

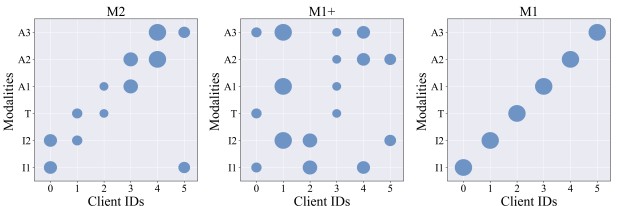

*Figure 3.* Example of dataset distribution partitioning.

**Baselines:** To ensure fair comparison, we re-implemented all baseline methods within a unified HtFLlib framework (Zhang et al., 2025a) and evaluated them under identical experimental settings. Specifically, we include representative prototype-based federated learning methods FedProto (Tan et al., 2022a), FedTGP (Zhang et al., 2024), and Fed-Pall (Zhang et al., 2025c), as well as state-of-the-art multimodal federated learning approaches Harmony (Ouyang et al., 2023), FedMVP (Che et al., 2024), and FedMobile (Liu et al., 2025), with local training (Local) serving as a reference baseline. For consistency, Local and all prototype-based methods adopt the same local multimodal aggregation strategy as our approach. Moreover, to accommodate model heterogeneity, methods that originally require uploading local models are uniformly modified to upload classifier

parameters instead. Details of all baselines are provided in Appendix C.3.

**Implementation details:** To ensure reliability, all methods are evaluated across five independent trials, and we report the mean performance alongside the standard deviation. SGD is adopted as the optimizer uniformly, with the number of local training epochs set to 2. Additionally, to accommodate varying dataset scales, the total number of global communication rounds is configured adaptively for each dataset. The details are provided in Appendix C.4.

### 6.2. Experimental Result

**Test Accuracy.** Table 2 reports the test accuracies of comparative methods on four multimodal datasets under the M2, M1+, and M1 partition settings. MFedPBA consistently achieves superior performance in all scenarios, with substantial improvements validating the efficacy of the dual-prototype bilateral alignment framework.

To further investigate feature space, we visualize the feature prototypes $E_k^m$ of three-modal clients in the Reuters dataset using t-SNE (Van der Maaten & Hinton, 2008), as shown in Figure 5. Despite sharing identical class labels, features extracted from different modalities are clearly separated due to heterogeneous encoders, corroborating our motivation on feature space incompatibility. Compared with Local training, our method exhibits more compact and well-structured clusters, indicating more effective representation learning. Moreover, in several settings, FedProto underperforms Local training, further confirming that naively aggregating misaligned prototypes under model heterogeneity can introduce knowledge contamination and lead to severe performance degradation rather than improvement.

Furthermore, we extended experiment to a larger number of clients to assess the scalability of the proposed framework. As summarized in Table 2 (K #Client number), the experimental results demonstrate that MFedPBA consistently maintains its superior performance even in large-scale FL settings. Detailed discussions and per-client performance results are provided in Appendix D.

**Communication Efficiency.** Theoretically, prototype-based FL significantly reduces communication overhead compared to methods that transmit full model parameters. The standard bidirectional communication cost for FedProto is $K2CM_kd_D$, our framework introduces only a marginal increase for transmitting logit prototypes, bringing the total complexity to $K2C(M_kd_D + d_C)$. Since the number of classes is typically several orders of magnitude smaller than the total model parameters, this additional overhead remains negligible. Consequently, our approach maintains high communication efficiency without imposing substantial costs relative to methods requiring parameters.

---

[4]http://archive.ics.uci.edu/ml/datasets.

*Table 2.* Comparison of average performance of different methods in simulations with different data distributions. M#: Data partition (K=M); K#: Number of large clients (in M1+). Best and second-best are **bold** and underlined respectively.

| Dataset | | Local | FedProto | FedTGP | FedPall | Harmony | FedMVP | FedMobile | **MFedPBA** |
|---|---|---|---|---|---|---|---|---|---|
| Caltech101 | M2 | $42.65_{\pm1.47}$ | $38.72_{\pm0.42}$ | $40.78_{\pm1.31}$ | $42.29_{\pm0.27}$ | $45.29_{\pm0.38}$ | $40.23_{\pm0.65}$ | $43.51_{\pm0.54}$ | $\mathbf{49.04_{\pm0.21}}$ |
| | M1+ | $40.83_{\pm1.12}$ | $36.81_{\pm0.35}$ | $40.71_{\pm1.23}$ | $41.8_{\pm0.77}$ | $43.04_{\pm0.64}$ | $38.91_{\pm1.25}$ | $42.29_{\pm0.6}$ | $\mathbf{46.82_{\pm0.54}}$ |
| | M1 | $38.47_{\pm0.45}$ | $31.57_{\pm1.23}$ | $36.18_{\pm0.72}$ | $40.77_{\pm0.36}$ | $40.1_{\pm0.69}$ | $38.92_{\pm0.62}$ | $40.09_{\pm0.13}$ | $\mathbf{43.64_{\pm0.11}}$ |
| | K50 | $29.38_{\pm1.63}$ | $25.17_{\pm1.78}$ | $28.77_{\pm1.43}$ | $30.59_{\pm1.26}$ | $30.58_{\pm0.65}$ | $27.74_{\pm1.5}$ | $30.88_{\pm0.81}$ | $\mathbf{31.19_{\pm0.92}}$ |
| Reuters | M2 | $70.61_{\pm0.28}$ | $73.39_{\pm0.57}$ | $69.73_{\pm0.88}$ | $73.08_{\pm1.04}$ | $72.18_{\pm0.41}$ | $74.07_{\pm1.08}$ | $73.19_{\pm0.26}$ | $\mathbf{75.16_{\pm0.45}}$ |
| | M1+ | $70.85_{\pm0.59}$ | $68.54_{\pm1.46}$ | $69.8_{\pm0.59}$ | $71.09_{\pm1.56}$ | $72.28_{\pm0.47}$ | $68.09_{\pm0.82}$ | $72.79_{\pm0.44}$ | $\mathbf{75.01_{\pm0.22}}$ |
| | M1 | $67.05_{\pm1.04}$ | $68.75_{\pm0.63}$ | $65.81_{\pm0.33}$ | $66.96_{\pm0.22}$ | $70.16_{\pm1.14}$ | $70.18_{\pm0.57}$ | $69.92_{\pm0.95}$ | $\mathbf{72.09_{\pm0.79}}$ |
| | K100 | $52.88_{\pm1.08}$ | $46.48_{\pm0.93}$ | $48.31_{\pm1.41}$ | $56.55_{\pm0.53}$ | $53.96_{\pm0.51}$ | $52.28_{\pm1.53}$ | $56.64_{\pm0.29}$ | $\mathbf{57.83_{\pm0.97}}$ |
| NUS-WIDE | M2 | $32.18_{\pm0.64}$ | $29.28_{\pm0.46}$ | $31.19_{\pm0.51}$ | $33.38_{\pm0.91}$ | $32.97_{\pm0.28}$ | $31.89_{\pm0.4}$ | $33.86_{\pm0.22}$ | $\mathbf{35.2_{\pm0.46}}$ |
| | M1+ | $28.72_{\pm0.71}$ | $27.79_{\pm0.55}$ | $27.13_{\pm0.18}$ | $30.99_{\pm0.63}$ | $31.56_{\pm0.61}$ | $29.82_{\pm0.59}$ | $30.95_{\pm0.57}$ | $\mathbf{32.15_{\pm0.36}}$ |
| | M1 | $31.15_{\pm0.14}$ | $24.96_{\pm0.52}$ | $29.55_{\pm0.73}$ | $31.02_{\pm0.96}$ | $32.19_{\pm0.35}$ | $32.7_{\pm0.51}$ | $31.2_{\pm0.36}$ | $\mathbf{33.25_{\pm0.24}}$ |
| | K20 | $24.52_{\pm0.5}$ | $24.47_{\pm0.95}$ | $24.15_{\pm0.37}$ | $26.35_{\pm0.44}$ | $26.28_{\pm0.41}$ | $26.61_{\pm0.86}$ | $27.34_{\pm0.37}$ | $\mathbf{28.08_{\pm0.37}}$ |
| Youtube | M2 | $41.72_{\pm0.66}$ | $41.12_{\pm0.68}$ | $45.16_{\pm1.02}$ | $44.23_{\pm0.3}$ | $43.8_{\pm0.76}$ | $45.53_{\pm0.52}$ | $44.83_{\pm0.53}$ | $\mathbf{47.92_{\pm0.93}}$ |
| | M1+ | $41.89_{\pm0.76}$ | $38.82_{\pm1.12}$ | $44.55_{\pm0.68}$ | $43.04_{\pm1.09}$ | $43.47_{\pm0.91}$ | $44.24_{\pm0.36}$ | $44.28_{\pm0.64}$ | $\mathbf{47.21_{\pm0.59}}$ |
| | M1 | $40.01_{\pm0.41}$ | $36.59_{\pm0.39}$ | $38.62_{\pm0.64}$ | $39.15_{\pm1.5}$ | $41.07_{\pm0.42}$ | $40.04_{\pm0.46}$ | $39.48_{\pm0.99}$ | $\mathbf{44.52_{\pm0.54}}$ |
| | K20 | $32.01_{\pm0.25}$ | $30.57_{\pm1.49}$ | $34.15_{\pm0.31}$ | $32.59_{\pm0.47}$ | $33.5_{\pm0.49}$ | $31.12_{\pm0.77}$ | $34.22_{\pm0.63}$ | $\mathbf{35.07_{\pm0.92}}$ |

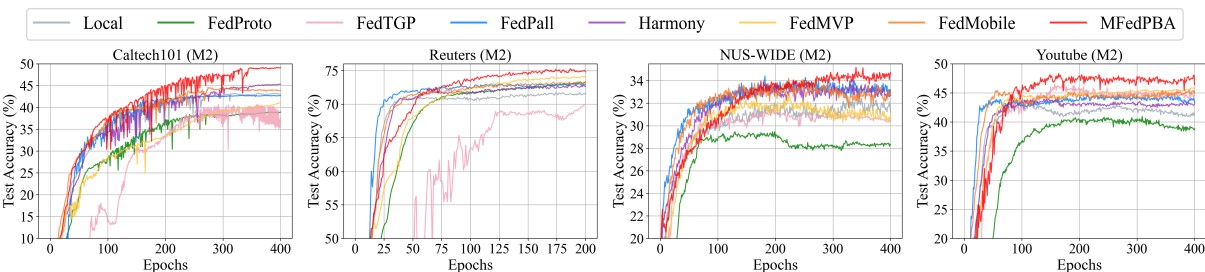

*Figure 4.* Test accuracy (%) on four datasets under the M2 data distribution setting with model heterogeneity.

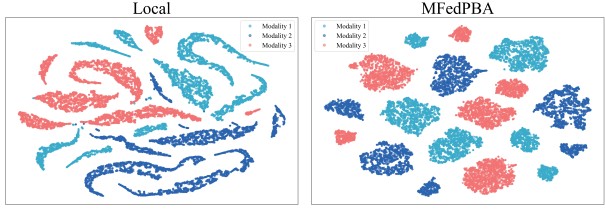

*Figure 5.* Feature visualization results of Local and MFedPBA.

We plotted the performance curves of various methods across four datasets under the M2 configuration, as shown in Figure 4, thereby confirming the stable convergence of our proposed framework. We observe that prototype-based baselines, particularly FedTGP, exhibit pronounced fluctuations during the training process. This instability stems from the erratic server prototype updates in multimodal scenarios, which hinder the acquisition of discriminative representations. In contrast, our feature prototypes are regularized by a multi-faceted objective function, allowing them to consolidate global semantic knowledge while preserving critical modality-specific discriminability.

**Computational Cost.** Compared to the baseline FedProto, the incremental client overhead is limited to the KL divergence between local and global logit prototypes, yielding a marginal complexity of $\mathcal{O}(C^2)$. On the server, logit aggregation follows a linear complexity of $\mathcal{O}(KCd_C)$, while feature-level operations encompass autoencoder mapping at $\mathcal{O}(\sum_k |\mathcal{M}_k|Cd_D^2)$, reconstruction loss calculation at $\mathcal{O}(\sum_k |\mathcal{M}_k|Cd_D)$, cross-modal contrastive loss at $\mathcal{O}(CM^2 d_D)$, and intra-modality structural alignment at $\mathcal{O}(\mathcal{T}_s MC^2)$, where $\mathcal{T}_s$ is the number of iterations of the Sinkhorn algorithm. Although these complexities scale with the class count $C$, they remain manageable within the scope of standard classification tasks. Crucially, our framework strategically offloads the primary computational burden to the resource-rich server, leveraging its superior processing power to facilitate high-precision alignment without straining resource-constrained clients. Figure 6 shows the running time of the client in each round on Reuters, further demonstrating that our computational overhead is acceptable.

**Ablation Study.** Figure 7 presents a comprehensive ablation study conducted under the Caltech101 M2 configuration to

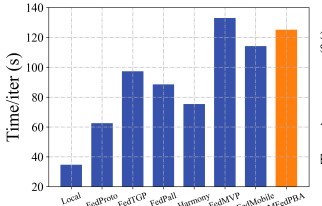

*Figure 6.* Time consumption.

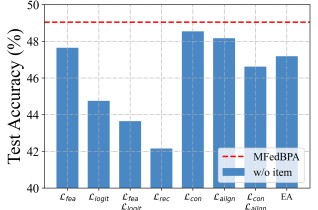

*Figure 7.* Ablation results.

evaluate the individual contributions of both server-side and client-side modules. In this analysis, the horizontal axis denotes the exclusion (w/o) of specific algorithmic components, while "EA" indicates the substitution of our proposed entropy-based weighting with standard equal averaging. The empirical results clearly demonstrate that the removal of any server-side module precipitates a significant drop in overall accuracy, thereby validating the necessity and synergy of the integrated design. Notably, eliminating the reconstruction loss ($\mathcal{L}_{rec}$) triggers the most severe performance degradation. This acute decline occurs because the absence of reconstruction constraints induces representation collapse within the latent space, which subsequently propagates erroneous global knowledge into the local training phase of individual clients.

**Impact of Feature Dimensions.** We systematically evaluate the impact of the feature dimension $d_D$ on model performance by varying it across a predefined set of values (e.g., $d_D \in \{24, 48, 64, 128, 256\}$), as summarized in Table 3. We observe a clear trade-off: smaller dimensions restrict the model's ability to capture complex data patterns, leading to underfitting, whereas excessively large dimensions introduce redundant noise and complicate classifier training. Specifically, on the YouTube dataset, most evaluated methods achieve their most stable and superior performance at $d_D = 48$. Based on this empirical observation, we independently conduct this sensitivity analysis across all datasets, ultimately selecting the dataset-specific optimal dimensions to strike the best balance between representational capacity and trainability.

*Table 3.* The test accuracy (%) on Youtube in the M1+ setting. "Fed" is omitted in the method name due to limited space.

| $d_D$ | Local | Proto | TGP | Pall | Harm | MVP | Mobile | PBA |
|---|---|---|---|---|---|---|---|---|
| 24-d | 41.19 | 30.12 | 41.19 | 42.56 | 42.06 | 43.85 | 40.91 | 44.43 |
| 48-d | 42.34 | 39.05 | 43.85 | 44.63 | 44.43 | 45.21 | 44.57 | 47.02 |
| 64-d | 43.02 | 37.82 | 45.22 | 42.27 | 43.13 | 43.21 | 44.24 | 46.46 |
| 128-d | 42.64 | 31.24 | 45.01 | 42.15 | 43.24 | 44.33 | 44.33 | 45.68 |
| 256-d | 41.95 | 26.32 | 43.22 | 41.75 | 42.74 | 42.84 | 42.54 | 43.84 |

**Hyper-parameter.** We analyze the sensitivity of server epochs $S$ and regularization terms $\lambda_1, \lambda_2$. As shown in Figure 8 (left). While accuracy improves with larger $S$, gains diminish significantly from 50 to 1000 epochs. Consequently,

we set $S = 10$ to balance efficiency and performance. We evaluated the sensitivity of the $\lambda_1, \lambda_2$ using Youtube in the range $\{0.001, 0.01, 0.05, 0.1, 0.5, 1, 5, 8, 10\}$. As shown in Figure 8 (right), MFedPBA demonstrates strong robustness across most parameter choices, consistently achieving accuracy above 46 and outperforming all baselines. Under several favorable configurations, the performance further improves to 48.9. In contrast, excessively large loss weights may introduce overly strong alignment constraints, leading to performance degradation.

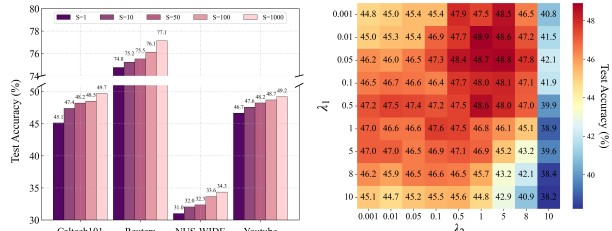

*Figure 8.* Hyperparameter experiments for $\lambda_1, \lambda_2$, and $S$.

# 7. Conclusion.

This paper addresses the challenges of model heterogeneity and modality imbalance in multimodal federated learning by proposing a prototype-guided bidirectional alignment framework for effective knowledge sharing across heterogeneous feature spaces. We analyze the limitations of conventional prototype aggregation under heterogeneous encoders and introduce a dual alignment mechanism that anchors knowledge transfer through semantically aligned logit prototypes while bridging embedding space discrepancies via GW–based feature alignment. Experimental results show that the proposed method effectively mitigates negative transfer and knowledge contamination, and significantly improves generalization and robustness under imbalanced and missing-modality settings.

## Acknowledgements

The research is supported by Seed Funding for Collaborative Research Grants of HKBU (with Grant No. RC-SFCRG/23-24/R2/SCI/06), the National Key Research and Development Program of China (2023YFB2703700), and the GMCC-SYSU Joint Lab for Smart Applications.

## Impact Statement

This paper presents work whose goal is to advance the field of Machine Learning. There are many potential societal consequences of our work, none which we feel must be specifically highlighted here.

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

## Summary of Appendix

In the appendix, the following contents are included:

## A. Algorithm

### A.1. Algorithm Framework

We summarize the main steps of the proposed in Algorithm 1.

### A.2. Privacy Analysis

Compared with federated learning paradigms that necessitate gradient or parameter sharing (Ouyang et al., 2023; Che et al., 2024; Liu et al., 2025), the proposed MFedPBA framework affords enhanced privacy preservation by constraining communication exclusively to feature and logit prototypes.

Feature prototypes are synthesized via client-specific non-linear encoders whose parameters remain strictly local. Consequently, the inverse reconstruction of raw inputs from embedding vectors presents a mathematically ill-posed problem, thereby significantly impeding data reconstruction attacks. Concurrently, logit prototypes abstract model outputs into class-level decision statistics, revealing minimal information regarding individual data instances. This abstraction renders it arduous for adversaries to distinguish whether a prototype is dominated by a solitary sensitive sample or derived from a diverse aggregate, effectively attenuating the efficacy of inference attacks.

Furthermore, the generation of both prototype modalities via class-wise averaging eliminates sample-level granularity, mitigating the risk of single-instance inference.

Finally, the server-side prototype optimization introduces an additional transformation layer, ensuring that the global prototypes distributed to clients are not direct replicas of any single client's representations, thus further reducing the probability of cross-client information leakage.

## B. Convergence Analysis

To analyze the convergence of MFedPBA, we follow (Tan et al., 2022a) and make the following assumptions B.1.

---

**Algorithm 1** MFedPBA

---

**Input**: total rounds $T$, local epochs $E$, server epochs $S$, total number of clients $K$, sampled number of clients $K_C$, local learning rate $\eta_c$, server learning rate $\eta_s$, hyper-parameter for loss $\lambda_1$ and $\lambda_2$.

**Server executes**:

  1: Initialize modality-specific encoders $\Phi_m(\cdot)$ and shared decoder $\psi(\cdot)$
  2: **for** each round $t = 1 \cdots T$ **do**
  3:     Server samples subset $K_C$ of clients
  4:     **for** each client $k \in K_C$ in parallel **do**
  5:         $\{\{E_{k,c}^m\}_{m=1}^{|\mathcal{M}_k|}, I_{k,c}\} \leftarrow$ **Client updates**$(\{\mathbf{P}_{G,c}^m\}, \mathbf{I}_{G,c})$
  6:     **end for**
  7:     Aggregate logit prototype by Eq. (6)
  8:     **for** each server epoch $s = 1 \cdots S$ **do**
  9:         Project client prototype to a unified space by Eq. (7)
10:         Aggregate modal feature prototype by Eq. (8)
11:         Calculate server loss update by Eq. (12)
12:     **end for**
13: **end for**

**Clients updates**:

  1: **for** each local epoch $e = 1 \cdots E$ **do**
  2:     Sample mini-batch in $\mathcal{B}$:
  3:     Calculate sample feature-based prototype $E_{k,c}^m$ by Eq. (2) and logit-based prototype $I_{k,c}$ by Eq. (3)
  4:     Calculate local loss by Eq. (16)
  5:     Update local model: $\theta_k^{t+1} \leftarrow \theta_k^t - \eta_c \nabla \mathcal{L}_k(\theta_k^t; \mathcal{B}_k)$
  6: **end for**
  7: **return** $\{E_{k,c}^m\}_{m=1}^{|\mathcal{M}_k|}$ and $I_{k,c}$

---

As for the iteration notation system, we define $t$ as the current communication round, $e \in \{0, 1, \cdots, E\}$ as the number of local iterations, where $E$ denotes the maximum number of local iterations. Thus, $(tE + e)$ represents the $e$-th iteration in the $(t + 1)$-th communication round. The $(t + 0)$ denotes that at the beginning of the $(t + 1)$-th round. Note that $(tE + E)$ corresponds to the last iteration in round $(t + 1)$. Moreover, $tE$ represents the time step before prototype aggregation, and $tE + 0$ represents the time step between prototype learning and the first iteration of the current round. Therefore, we can decompose one round of communication into two stages:
(1) Local update phase: $[tE + 0 \rightarrow (t + 1)E]$ indicates that the client has completed the local update.
(2) Server learning phase: $[(t + 1)E \rightarrow (t + 1)E + 0]$ indicates that the server updates the prototype and distributes it.

Here, we provide detailed mathematical expressions to better represent the process of updating local models. We split each client $k$'s model $\theta_k$ into a feature extractor $f_k^m$ parameterized by $\varphi_k$ and a classifier $h_k$ parameterized by $w_k$. Each client is equipped with $|\mathcal{M}_k|$ modality specific feature extractors, denoted as $f_k^m : \mathcal{X}_{\mathcal{M}_k} \rightarrow \mathbb{R}^{d_D}$. The client possesses a single classifier, denoted as $h_k : \mathbb{R}^{d_D} \rightarrow \mathbb{R}^{d_C}$. So the client loss function can be written as $\mathcal{F}_k = \{f_k^m(\varphi_k^m)\}_m \circ h_k(w_k)$, and sometimes we use $\theta_k$ to represent $(\{\varphi_k^m\}_m, w_k)$ for short. Therefore, the local loss function of client $k$ can be written as:

$$\mathcal{L}(\theta_k; \boldsymbol{x}, y) = \mathcal{L}_{ce}(\mathcal{F}_k(\theta_k; \boldsymbol{x}), y) + \lambda_1 \sum_m ||f_k^m(\varphi_k^m; \boldsymbol{x}^m) - \mathbf{P}_{G,c}^m||_2^2 + \lambda_2 ||h_k(w_k; f_k^m \boldsymbol{x}^m) - \mathbf{I}_{G,c}||_2^2. \tag{21}$$

It is worth noting that $\lambda_2$ serves as the weighting coefficient for the KL divergence term. Since the KL divergence can be approximated by the quadratic Mean Squared Error (MSE) loss via second-order Taylor expansion near the convergence point, we formulate the objective in Eq. (21) using the MSE form to facilitate the theoretical convergence analysis.

Moreover, the global feature prototype $\mathbf{P}_{G,c}^m$ is obtained by the server via $S$ rounds of training with the loss function $\mathcal{L}_{\text{server}}$, while the global logit prototype $\mathbf{I}_{G,c}$ is aggregated at the server. Therefore, the server optimizes the global loss for $S$ rounds can be written as:

$$\mathcal{L}_{\text{server}}(\mathbf{P}) = \mathcal{L}_{\text{rec}} + \mathcal{L}_{\text{con}} + \mathcal{L}_{\text{align}} = \frac{1}{K} \sum_{k,m,c} ||P_{k,c}^m - E_{k,c}^m||^2 + \mathcal{L}_{\text{con}}(\mathbf{P}) + \mathcal{L}_{\text{align}}(\mathbf{P}, \mathbf{I}). \tag{22}$$

For ease of analysis, we will represent both loss $\mathcal{L}_{\text{con}}$ and $\mathcal{L}_{\text{align}}$ together as $\mathcal{L}_{\text{m}}$, i.e., $\mathcal{L}_{\text{m}} = \mathcal{L}_{\text{con}} + \mathcal{L}_{\text{align}}$.

## B.1. Assumption

**Assumption B.1** (Lipschitz Smoothness). The $k$-th client's local model loss function $\mathcal{L}$ is Lipschitz continuous with Lipschitz constant $L$, and $L > 0$ with $\mathcal{L}(0) = 0$, i.e.,

$$\|\nabla\mathcal{L}_{t_1} - \nabla\mathcal{L}_{t_2}\|_2 \le L\|\theta_{k,t_1} - \theta_{k,t_2}\|_2, \quad \forall t_1, t_2 > 0, \quad k \in \{1, 2, \ldots, K\}. \tag{23}$$

which implies the following quadratic bound,

$$\mathcal{L}_{t_1} - \mathcal{L}_{t_2} \le \langle \nabla\mathcal{L}_{t_2}, (\theta_{k,t1} - \theta_{k,t2}) \rangle + \frac{L}{2}\|\theta_{k,t1} - \theta_{k,t2}\|_2^2, \quad \forall t_1, t_2 > 0, \quad k \in \{1, 2, \ldots, K\}. \tag{24}$$

**Assumption B.2** (Unbiased Gradient and Bounded Variance). The random gradient $g_{k,t} = \nabla\mathcal{L}_t(\theta_{k,t}; \mathcal{B}_{k,t})$ of each client's local model is unbiased, where $\mathcal{B}$ is a batch of local data, i.e.,

$$\mathbb{E}_{\mathcal{B}_{k,t} \subseteq N_k}[g_{k,t}] = \nabla\mathcal{L}(\theta_{k,t}) = \nabla\mathcal{L}_t, \quad \forall k \in \{1, 2, \ldots, K\}, \tag{25}$$

and the variance of random gradient $g_{k,t}$ is bounded by:

$$\mathbb{E}_{\mathcal{B}_{k,t} \subseteq N_k}\left[\|\nabla\mathcal{L}_t(\theta_{k,t}; \mathcal{B}_{k,t}) - \nabla\mathcal{L}_t(\theta_{k,t})\|_2^2\right] \le \sigma^2, \quad \forall k \in \{1, 2, \ldots, K\}. \tag{26}$$

**Assumption B.3** (Bounded Gradients). The expectation of the stochastic gradient is bounded by $G_c$:

$$\mathbb{E}\left[\|\nabla\mathcal{L}_k\|^2\right] \le G_c^2, \quad \forall k \in \{1, 2, \ldots, K\}. \tag{27}$$

**Assumption B.4** (Lipschitz Continuity). For client $k$, feature extraction $f_k^m(\varphi_k^m)$ and classifier $h_k(w_k)$ are Lipschitz continuous with Lipschitz constant $L_r$, and $L_r > 0$:

$$\begin{aligned}
\|f_k^m(\varphi_{k,t_1}^m) - f_k^m(\varphi_{k,t_2}^m)\|_2 &\le L_r\|\varphi_{k,t_1}^m - \varphi_{k,t_2}^m\|_2, \\
\|h_k(w_{k,t_1}) - \nabla h_k(w_{k,t_2})\|_2 &\le L_r\|w_{k,t_1} - w_{k,t_2}\|_2.
\end{aligned} \tag{28}$$

**Assumption B.5** (Bounded Server Optimization Drift). The server objective function $\mathcal{L}_{\text{server}}$ is $L_s$-smooth, and the expectation of the stochastic gradient is bounded, i.e.,

*server Lipschitz Smoothness*: $\|\nabla\mathcal{L}_{\text{server},t_1} - \nabla\mathcal{L}_{\text{server},t_2}\|_2 \le L_s\|\theta_{k,t_1} - \theta_{k,t_2}\|_2$,

*gradient bounded*: $\mathbb{E}\left[\|\nabla\mathcal{L}_{\text{m}}\|^2\right] \le G_s^2$, and $\mathbb{E}\left[\|\nabla\mathcal{L}_{\text{server}}\|^2\right] \le \delta_s^2$.

## B.2. Lemmas & Theorems

Based on the above assumptions, MFedPBA uses a prototype-based approach for local training updates, Lemma B.6 derived by Tan *et al.,* (Tan et al., 2022a) still holds.

**Lemma B.6.** *Let Assumption B.1 and B.2 hold. From the beginning of communication round $t + 1$ to the last local update step, the loss function of an arbitrary client can be bounded as:*

$$\mathbb{E}[\mathcal{L}_{(t+1)E}] \le \mathcal{L}_{tE+0} - (\eta_c - \frac{L\eta_c^2}{2})\sum_{e=0}^{E}\|\nabla\mathcal{L}_{tE+e}\|_2^2 + \frac{LE\eta_c^2}{2}\sigma^2. \tag{29}$$

**Lemma B.7.** *Let Assumption B.3, B.4 and B.5 hold. After feature prototype learning and logit prototype aggregation are completed on the server, the loss function of any client can be constrained as follows:*

$$\mathbb{E}[\mathcal{L}_{(t+1)E+0}] \le \mathcal{L}_{(t+1)E} + (M\lambda_1 + \lambda_2)L_r E\eta_c G_c + 2\lambda_1(\delta_s + MG_s). \tag{30}$$

**Theorem B.8** (One-round deviation). *Based on the above assumptions, using $\Gamma_{\text{d}} = (M\lambda_1 + \lambda_2)L_r G_c$ and $\Gamma_{\text{s}} = 2\lambda_1(\delta_s + MG_s)$ represent the drift constant and server bias constant. The expectation of the loss of an arbitrary client's local model before the start of a round of local iteration satisfies:*

$$\mathbb{E}[\mathcal{L}_{(t+1)E+0}] \le \mathcal{L}_{tE+0} - (\eta_c - \frac{L\eta_c^2}{2})\sum_{e=0}^{E}\|\nabla\mathcal{L}_{tE+e}\|_2^2 + \frac{LE\eta_c^2}{2}\sigma^2 + \Gamma_d E\eta_c + \Gamma_s. \tag{31}$$

**Theorem B.9** (Non-convex convergence rate of MFedPBA)**.** *Based on the above assumptions, for an arbitrary client and any $\epsilon > 0$, the following inequality holds:*

$$\frac{1}{T}\sum_{t=0}^{T-1}\sum_{e=0}^{E}\mathbb{E}\left[\|\mathcal{L}_{tE+e}\|_2^2\right] \leq \frac{2\left(\mathcal{L}_{t=1} - \mathcal{L}^*\right)}{T\eta_c\left(2 - L\eta_c\right)} + \frac{LE\eta_c^2\sigma^2 + 2(\Gamma_d E\eta_c + \Gamma_s)}{2\eta_c - L\eta_c^2},$$

$$\leq \epsilon. \tag{32}$$

$$s.t. \quad \eta_c < \min\left\{\frac{2}{L}, \frac{(\epsilon - \Gamma_d E) + \sqrt{(\epsilon - \Gamma_d E)^2 - 2L(\epsilon + E\sigma^2)\Gamma_s}}{L(\epsilon + E\sigma^2)}\right\}.$$

### B.3. Proof of Lemma 5.1

**Lemma 5.1.** Let Assumption B.1 and B.2 hold. From the beginning of communication round $t + 1$ to the last local update step, the loss function of an arbitrary client can be bounded as:

$$\mathbb{E}[\mathcal{L}_{(t+1)E}] \leq \mathcal{L}_{tE+0} - (\eta_c - \frac{L\eta_c^2}{2})\sum_{e=0}^{E}\|\nabla\mathcal{L}_{tE+e}\|_2^2 + \frac{LE\eta_c^2}{2}\sigma^2.$$

*Proof.* For arbitrary clients, we have $\theta_{t+1} = \theta_t - \eta_c g_t$, then

$$\mathcal{L}_{tE+1} \leq \mathcal{L}_{tE+0} + \langle\nabla\mathcal{L}_{tE+0}, (\theta_{tE+1} - \theta_{tE+0})\rangle + \frac{L}{2}\|\theta_{tE+1} - \theta_{tE+0}\|_2^2$$

$$= \mathcal{L}_{tE+0} - \eta_c\langle\nabla\mathcal{L}_{tE+0}, g_{tE+0}\rangle + \frac{L}{2}\|\eta_c g_{tE+0}\|_2^2. \tag{33}$$

Taking expectation of both sides of the above equation on the random variable $\mathcal{B}$, we have

$$\mathbb{E}[\mathcal{L}_{tE+1}] \leq \mathcal{L}_{tE+0} - \eta_c\mathbb{E}[\langle\nabla\mathcal{L}_{tE+0}, g_{tE+0}\rangle] + \frac{L\eta_c^2}{2}\mathbb{E}[\|g_{tE+0}\|_2^2]$$

$$= \mathcal{L}_{tE+0} - \eta_c\|\nabla\mathcal{L}_{tE+0}\|_2^2 + \frac{L\eta_c^2}{2}\mathbb{E}[\|g_{k,tE+0}\|_2^2]$$

$$\leq \mathcal{L}_{tE+0} - \eta_c\|\nabla\mathcal{L}_{tE+0}\|_2^2 + \frac{L\eta_c^2}{2}(\|\nabla\mathcal{L}_{tE+0}\|_2^2 + \mathrm{Var}(g_{k,tE+0})) \tag{34}$$

$$= \mathcal{L}_{tE+0} - (\eta_c - \frac{L\eta_c^2}{2})\|\nabla\mathcal{L}_{tE+0}\|_2^2 + \frac{L\eta_c^2}{2}\mathrm{Var}(g_{k,tE+0})$$

$$\leq \mathcal{L}_{tE+0} - (\eta_c - \frac{L\eta_c^2}{2})\|\nabla\mathcal{L}_{tE+0}\|_2^2 + \frac{L\eta_c^2}{2}\sigma^2,$$

where $\mathrm{Var}(x) = \mathbb{E}[x^2] - (\mathbb{E}[x])^2$. Take expectation of $\theta$ on both sides. Then, by telescoping of $E$ steps, we have,

$$\mathbb{E}[\mathcal{L}_{(t+1)E}] \leq \mathcal{L}_{tE+0} - (\eta_c - \frac{L\eta_c^2}{2})\sum_{e=0}^{E}\|\nabla\mathcal{L}_{tE+e}\|_2^2 + \frac{LE\eta_c^2}{2}\sigma^2, \tag{35}$$

which completes the proof. □

### B.4. Proof of Lemma 5.2

**Lemma 5.2.** Let Assumption B.3, B.4 and B.5 hold. After feature prototype learning and logit prototype aggregation are completed on the server, the loss function of any client can be constrained as follows:

$$\mathbb{E}[\mathcal{L}_{(t+1)E+0}] \leq \mathcal{L}_{(t+1)E} + (M\lambda_1 + \lambda_2)L_r E\eta_c G_c + 2\lambda_1(\delta_s + MG_s).$$

*Proof.* We define the temporal states of a client. Time $(t + 1)E$ denotes the moment when the client has just completed $E$ local training epochs in the $t$-th round but has not yet received the new prototypes from the server. Time $(t + 1)E + 0$ denotes the moment when the client receives the updated prototypes dispatched by the server.

First, we have: $\mathcal{L}_{(t+1)E+0} = \mathcal{L}_{(t+1)E} + \mathcal{L}_{(t+1)E+0} - \mathcal{L}_{(t+1)E}$. Therefore, our main goal is to derive an upper bound for the difference in loss before and after aggregation:

$$\Delta\mathcal{L} = \mathbb{E}[\mathcal{L}_{(t+1)E+0} - \mathcal{L}_{(t+1)E}]. \tag{36}$$

Since the model parameter $\theta$ does not change during aggregation, the cross-entropy loss $\mathcal{L}_{ce}$ cancels out. The difference only comes from the changes in the global prototypes in the regularization term. Thus we have:

$$\begin{aligned}
\mathcal{L}_{(t+1)E+0} - \mathcal{L}_{(t+1)E} =& \lambda_1 \sum_m \left( ||f_k^m(\phi_{k,(t+1)E}^m) - \mathbf{P}_{G,t+2}^m|| - ||f_k^m(\phi_{k,(t+1)E}^m) - \mathbf{P}_{G,t+1}^m|| \right) \\
&+ \lambda_2 \left( ||h_k(w_{k,(t+1)E}) - \mathbf{I}_{G,t+2}|| - ||h_k(w_{k,(t+1)E}) - \mathbf{I}_{G,t+1}|| \right).
\end{aligned} \tag{37}$$

Using the reverse triangle inequality $\|a - b\|_2 - \|a - c\|_2 \le \|b - c\|_2$, we can bound the above expression as follows:

$$\mathcal{L}_{(t+1)E+0} - \mathcal{L}_{(t+1)E} = \underbrace{\lambda_1 \sum_m \|\mathbf{P}_{G,t+2}^m - \mathbf{P}_{G,t+1}^m\|}_{A} + \underbrace{\lambda_2 \|\mathbf{I}_{G,t+2} - \mathbf{I}_{G,t+1}\|}_{B}. \tag{38}$$

Next, we define the changes in Part A and Part B separately.

Since the logical value prototype of Part B uses weighted aggregation, we will begin our analysis with Part B. From Eq. (6), we can see that:

$$\mathbf{I}_{G,t+2} = \sum_{k=1}^K q_k I_{k,(t+1)E}, \quad \mathbf{I}_{G,t+1} = \sum_{k=1}^K q_k I_{k,tE}. \tag{39}$$

Therefore, we have:

$$\begin{aligned}
\|\mathbf{I}_{G,t+2} - \mathbf{I}_{G,t+1}\| &= \left\| \sum_{k=1}^K q_k I_{k,(t+1)E} - \sum_{k=1}^K q_k I_{k,tE} \right\|_2 = \left\| \sum_{k=1}^K q_k (I_{k,(t+1)E} - I_{k,tE}) \right\|_2 \\
&\le \left\| \sum_{k=1}^K q_k \frac{1}{N_k} \sum_{i=1}^{N_k} (h_k(w_{k,(t+1)E}; \boldsymbol{x}_{k,i}) - h_k(w_{k,tE}; \boldsymbol{x}_{k,i})) \right\|_2 \\
&\le \sum_{k=1}^K q_k \frac{1}{N_k} \sum_{i=1}^{N_k} \left\| h_k(w_{k,(t+1)E}; \boldsymbol{x}_{k,i}) - h_k(w_{k,t+E}; \boldsymbol{x}_{k,i}) \right\|_2 \\
&\le L_r \sum_{k=1}^K q_k \left\| w_{k,(t+1)E} - w_{k,tE} \right\|_2 \\
&\le L_r \sum_{k=1}^K q_k \left\| \theta_{k,(t+1)E} - \theta_{k,tE} \right\|_2 = L_r \eta_c \sum_{k=1}^K q_k \left\| \sum_{e=0}^{E-1} g_{k,tE+e} \right\|_2 \\
&\le L_r \eta_c \sum_{k=1}^K q_k \sum_{e=0}^{E-1} \|g_{k,tE+e}\|_2.
\end{aligned} \tag{40}$$

Take expectations on both sides, since $\sum q_k = 1$, then:

$$\lambda_2 \|\mathbf{I}_{G,t+2} - \mathbf{I}_{G,t+1}\| \le \lambda_2 L_r \eta_c \sum_{k=1}^K q_k \sum_{e=0}^{E-1} \|g_{k,tE+e}\|_2 \le \lambda_2 L_r E \eta_c G_c. \tag{41}$$

The derivation for Part A is complete.

Since the global feature prototype is aggregated using non-convex SGD, the server prototype $\mathbf{P}_G^m$ is the result of the client uploading and optimizing $E_k^m$ for $S$ rounds. Based on Assumption B.5, we introduce the aggregated mean as an intermediate variable.

Let $\mathbf{E}_{t+2} = \sum q_k E_{k,t+2}$ be the geometric center of the currently uploaded feature prototypes, and $\mathbf{E}_{t+1} = \sum q_k E_{k,t+1}$ be the geometric center of the features uploaded in the previous round. Here, we omit the modality superscript $m$ for ease of analysis. Therefore, for part A, we have:

$$
\begin{aligned}
\|\mathbf{P}_{G,t+2} - \mathbf{P}_{G,t+1}\| &= \|\mathbf{P}_{G,t+2} - \mathbf{E}_{t+2} + \mathbf{E}_{t+2} - \mathbf{E}_{t+1} + \mathbf{E}_{t+1} - \mathbf{P}_{G,t+1}\| \\
&\leq \underbrace{\|\mathbf{P}_{G,t+2} - \mathbf{E}_{t+2}\|}_{A1} + \underbrace{\|\mathbf{E}_{t+2} - \mathbf{E}_{t+1}\|}_{A2} + \underbrace{\|\mathbf{E}_{t+1} - \mathbf{P}_{G,t+1}\|}_{A3}.
\end{aligned}
\tag{42}
$$

We found that Part A2 is completely consistent with the derivation of the logical value prototype, and the change in the mean is limited by the drift of the client's local parameters. Therefore, we have:

$$
\|\mathbf{E}_{t+2} - \mathbf{E}_{t+1}\| \leq L_r E \eta_c G_c.
\tag{43}
$$

By solving for the gradient of Eq. (22), we obtain:

$$
\begin{aligned}
\nabla \mathcal{L}_{server}(\mathbf{P}_G^m) &= \frac{1}{2K} \sum_{k=1}^{K} \sum_{m=1}^{M} 2(P_k^m - E_k^m) + \sum_{m=1}^{M} \nabla \mathcal{L}_m(\mathbf{P}_G^m) \\
&= M(\mathbf{P}_G - \mathbf{E}_{avg}) + M \nabla \mathcal{L}_m(\mathbf{P}_G^m).
\end{aligned}
\tag{44}
$$

Through algebraic manipulation, we obtain:

$$
\mathbf{P}_G - \mathbf{E}_{avg} = \frac{1}{M} \nabla \mathcal{L}_{\text{server}}(\mathbf{P}_G^m) - \nabla \mathcal{L}_m(\mathbf{P}_G^m).
\tag{45}
$$

By applying the L2-norm and the triangle inequality to the above expression, we obtain:

$$
\|\mathbf{P}_G - \mathbf{E}_{avg}\|_2 \leq \|\frac{1}{M} \nabla \mathcal{L}_{\text{server}}(\mathbf{P}_G^m)\|_2 + \|\nabla \mathcal{L}_m(\mathbf{P}_G^m)\|_2.
\tag{46}
$$

According to assumption B.5, therefore we have:

$$
\begin{aligned}
\mathbb{E}[\|\mathbf{P}_G - \mathbf{E}_{avg}\|_2] &\leq \mathbb{E}[\|\frac{1}{M} \nabla \mathcal{L}_{\text{server}}(\mathbf{P}_G^m)\|_2] + \mathbb{E}[\|\nabla \mathcal{L}_m(\mathbf{P}_G^m)\|_2] \\
&\leq \frac{\delta_s}{M} + G_s.
\end{aligned}
\tag{47}
$$

Accordingly, the derivation process for A1 and A3 is the same as described above. Therefore, considering that there are M modes in total, we can integrate the above process into Eq. (42) to obtain:

$$
\begin{aligned}
\|\mathbf{P}_{G,t+2} - \mathbf{P}_{G,t+1}\| &\leq \underbrace{\|\mathbf{P}_{G,t+2} - \mathbf{E}_{t+2}\|}_{A1} + \underbrace{\|\mathbf{E}_{t+2} - \mathbf{E}_{t+1}\|}_{A2} + \underbrace{\|\mathbf{E}_{t+1} - \mathbf{P}_{G,t+1}\|}_{A3}. \\
&\leq 2M(\frac{\delta_s}{M} + G_s) + M L_r E \eta_c G_c = 2\delta_s + 2M G_s + M L_r E \eta_c G_c.
\end{aligned}
\tag{48}
$$

Based on the derivation results from Eq. (48) and Eq. (41), substituting them into Eq. (38), and taking expectation of both sides:

$$
\begin{aligned}
\mathbb{E}[\mathcal{L}_{(t+1)E+0}] - \mathbb{E}[\mathcal{L}_{(t+1)E}] &\leq \lambda_1(2\delta_s + 2M G_s + M L_r E \eta_c G_c) + \lambda_2 L_r E \eta_c G_c \\
&= (M\lambda_1 + \lambda_2) L_r E \eta_c G_c + 2\lambda_1(\delta_s + M G_s),
\end{aligned}
\tag{49}
$$

which completes the proof. $\qquad\square$

### B.5. Proof of Theorem 5.3

**Theorem 5.3** Based on the above assumptions, using $\Gamma_d = (M\lambda_1 + \lambda_2) L_r G_c$ and $\Gamma_s = 2\lambda_1(\delta_s + M G_s)$ represent the drift constant and server bias constant. The expectation of the loss of an arbitrary client's local model before the start of a round of local iteration satisfies:

$$
\begin{aligned}
\mathbb{E}[\mathcal{L}_{(t+1)E+0}] &\leq \mathcal{L}_{tE+0} - (\eta_c - \frac{L\eta_c^2}{2}) \sum_{e=0}^{E} \|\nabla \mathcal{L}_{tE+e}\|_2^2 + \frac{LE\eta_c^2}{2}\sigma^2 \\
&\quad + \Gamma_d E \eta_c + \Gamma_s.
\end{aligned}
$$

*Proof.* According to the expectation equation, the change in one communication round consists of two phases: the server-side phase and the client-side phase. The total change can be expressed as:

$$\mathbb{E}[\mathcal{L}_{(t+1)E+0}] - \mathcal{L}_{tE+0} \leq \underbrace{\left(\mathbb{E}[\mathcal{L}_{(t+1)E+0}] - \mathbb{E}[\mathcal{L}_{(t+1)E}]\right)}_{\text{server}} + \underbrace{\left(\mathbb{E}[\mathcal{L}_{(t+1)E}] - \mathbb{E}[\mathcal{L}_{tE+0}]\right)}_{\text{client}}. \tag{50}$$

Substituting Lemma B.6 into the second term on the right-hand side of Lemma B.7, we have:

$$\begin{aligned}
\mathbb{E}[\mathcal{L}_{(t+1)E+0}] &\leq \mathbb{E}[\mathcal{L}_{(t+1)E}] + (M\lambda_1 + \lambda_2)L_r E\eta_c G_c + 2\lambda_1(\delta_s + MG_s) \\
&\leq \mathcal{L}_{tE+0} - (\eta_c - \frac{L\eta_c^2}{2})\sum_{e=0}^{E}\|\nabla\mathcal{L}_{tE+e}\|_2^2 + \frac{LE\eta_c^2}{2}\sigma^2 \\
&\quad + (M\lambda_1 + \lambda_2)L_r E\eta_c G_c + 2\lambda_1(\delta_s + MG_s).
\end{aligned} \tag{51}$$

To simplify the notation, we let $\Gamma_d = (M\lambda_1 + \lambda_2)L_r G_c$ represent the drift constant, and $\Gamma_s = 2\lambda_1(\delta_s + MG_s)$ represent the server bias constant, then we have

$$\begin{aligned}
\mathbb{E}[\mathcal{L}_{(t+1)E+0}] &\leq \mathcal{L}_{tE+0} - (\eta_c - \frac{L\eta_c^2}{2})\sum_{e=0}^{E}\|\nabla\mathcal{L}_{tE+e}\|_2^2 + \frac{LE\eta_c^2}{2}\sigma^2 \\
&\quad + \Gamma_d E\eta_c + \Gamma_s,
\end{aligned} \tag{52}$$

which completes the proof. $\qquad\square$

### B.6. Proof of Theorem 5.4

**Theorem 5.4** Based on the above assumptions, for an arbitrary client and any $\epsilon > 0$, the following inequality holds:

$$\begin{aligned}
\frac{1}{T}\sum_{t=0}^{T-1}\sum_{e=0}^{E}\mathbb{E}\left[\|\mathcal{L}_{tE+e}\|_2^2\right] &\leq \frac{2(\mathcal{L}_{t=1} - \mathcal{L}^*)}{T\eta_c(2 - L\eta_c)} + \frac{LE\eta_c^2\sigma^2 + 2(\Gamma_d E\eta_c + \Gamma_s)}{2\eta_c - L\eta_c^2} \\
&\leq \epsilon. \\
s.t. \quad \eta_c &< \min\left\{\frac{2}{L}, \frac{(\epsilon - \Gamma_d E) + \sqrt{(\epsilon - \Gamma_d E)^2 - 2L(\epsilon + E\sigma^2)\Gamma_s}}{L(\epsilon + E\sigma^2)}\right\}.
\end{aligned}$$

*Proof.* Transform the form of Theorem B.8 into

$$\sum_{e=0}^{E}\|\mathcal{L}_{tE+e}\|_2^2 \leq \frac{\mathcal{L}_{tE+0} - \mathbb{E}\left[\mathcal{L}_{(t+1)E+0}\right] + \frac{LE\eta_c^2}{2}\sigma^2 + \Gamma_d E\eta_c + \Gamma_s}{\eta_c - \frac{L\eta_c^2}{2}}. \tag{53}$$

Take expectations of model $\theta$ on both sides, we have:

$$\sum_{e=0}^{E}\mathbb{E}\left[\|\mathcal{L}_{tE+e}\|_2^2\right] \leq \frac{\mathbb{E}\left[\mathcal{L}_{tE+0}\right] - \mathbb{E}\left[\mathcal{L}_{(t+1)E+0}\right] + \frac{LE\eta_c^2}{2}\sigma^2 + \Gamma_d E\eta_c + \Gamma_s}{\eta_c - \frac{L\eta_c^2}{2}}. \tag{54}$$

Summing both sides of Eq. (54) over $T$ rounds, since $\sum_{t=1}^{T}\left(\mathbb{E}\left[\mathcal{L}_{tE+0}\right] - \mathbb{E}\left[\mathcal{L}_{(t+1)E+0}\right]\right) \leq \mathcal{L}_{t=0} - \mathcal{L}^*$, for each round:

$$\begin{aligned}
\frac{1}{T}\sum_{t=0}^{T-1}\sum_{e=0}^{E}\mathbb{E}\left[\|\mathcal{L}_{tE+e}\|_2^2\right] &\leq \frac{\frac{1}{T}\sum_{t=0}^{T-1}\left(\mathbb{E}\left[\mathcal{L}_{tE+0}\right] - \mathbb{E}\left[\mathcal{L}_{(t+1)E+0}\right]\right) + \frac{LE\eta_c^2}{2}\sigma^2 + \Gamma_d E\eta_c + \Gamma_s}{\eta_c - \frac{K\eta_c^2}{2}} \\
&\leq \frac{\frac{1}{T}(\mathcal{L}_{t=0} - \mathcal{L}^*) + \frac{LE\eta_c^2}{2}\sigma^2 + \Gamma_d E\eta_c + \Gamma_s}{\eta_c - \frac{L\eta_c^2}{2}} \\
&= \frac{2(\mathcal{L}_{t=0} - \mathcal{L}^*) + TLE\eta_c^2\sigma^2 + 2T(\Gamma_d E\eta_c + \Gamma_s)}{T(2\eta_c - L\eta_c^2)} \\
&= \frac{2(\mathcal{L}_{t=0} - \mathcal{L}^*)}{T\eta_c(2 - L\eta_c)} + \frac{LE\eta_c^2\sigma^2 + 2(\Gamma_d E\eta_c + \Gamma_s)}{2\eta_c - L\eta_c^2}.
\end{aligned} \tag{55}$$

Given any $\epsilon > 0$ the above equation satisfies

$$\frac{2\left(\mathcal{L}_{t=0} - \mathcal{L}^*\right)}{T\eta_c\left(2 - L\eta_c\right)} + \frac{LE\eta_c^2\sigma^2 + 2(\Gamma_d E\eta_c + \Gamma_s)}{2\eta_c - L\eta_c^2} \leq \epsilon. \tag{56}$$

Then, we can obtain:

$$T \geq \frac{2\left(\mathcal{L}_{t=0} - \mathcal{L}^*\right)}{\eta_c\epsilon\left(2 - L\eta_c\right) - \eta_c E(L\eta_c\sigma^2 + 2\Gamma_d\eta_c) - 2\Gamma_s}. \tag{57}$$

Since $T > 0, \mathcal{L}_{t=0} - \mathcal{L}^* > 0$, we can further derive:

$$\eta_c\epsilon\left(2 - L\eta_c\right) - \eta_c E(L\eta_c\sigma^2 + 2\Gamma_d\eta_c) - 2\Gamma_s > 0, \tag{58}$$

Since Eq. (58) is a quadratic function that opens upwards, we solve it to obtain:

$$\eta_c < \min\left\{\frac{2}{L}, \frac{(\epsilon - \Gamma_d E) + \sqrt{(\epsilon - \Gamma_d E)^2 - 2L(\epsilon + E\sigma^2)\Gamma_s}}{L(\epsilon + E\sigma^2)}\right\}, \tag{59}$$

When $\Gamma_s$ approaches 0, Eq. (59) can be simplified to:

$$\eta_c < \min\left\{\frac{2}{L}, \frac{2(\epsilon - \Gamma_d E)}{L(\epsilon + E\sigma^2)}\right\}, \tag{60}$$

which completes the proof. $\qquad\square$

## C. Details of the Experimental Setup

### C.1. Dataset Description

This paper focuses on multimodal federated learning under model heterogeneity and imbalanced modality distributions. To evaluate performance in modality-imbalanced scenarios, we select multiple datasets with diverse modalities and classes and construct distributed federated settings, denoted as datasets Caltech101 [5], Reuters [6], NUS-WIDE [7], and Youtube [8]. Detailed descriptions of each dataset are provided below.

The **Caltech101 dataset** is a widely used benchmark dataset introduced by the California Institute of Technology in 2003 and has been extensively adopted in computer vision, pattern recognition, and machine learning research. The dataset is designed to provide a well-annotated and low-noise image collection for object recognition and image classification tasks. It consists of 102 categories, including 101 object classes and one background category (BACKGROUND_Google). To fully exploit the visual information in Caltech101, we adopt a multimodal feature extraction strategy that characterizes images from complementary perspectives. Specifically, we consider the following three modalities:

- I1 (Histogram of Oriented Gradients): A 1984-dimensional Histogram of Oriented Gradients feature, which captures local structural and edge information. Images are partitioned into connected regions, within which gradient orientations are computed and aggregated into histograms. The regional histograms are then concatenated and block-normalized to improve robustness to illumination variations and local deformations.

- I2 (Gist Descriptor): A 512-dimensional Gabor feature designed to model texture and frequency responses. A bank of multi-scale and multi-orientation Gabor filters is applied to each image, followed by average pooling over a 4×4 spatial grid. The pooled responses from all filters and grid cells are concatenated to form the final representation.

- I3 (Local Binary Pattern): A 928-dimensional Local Binary Pattern feature that encodes local texture patterns. For each pixel, binary codes are generated by comparing the center pixel with its neighbors and converted into decimal values. The global image representation is obtained by computing the histogram of LBP codes over the entire image.

---

[5]https://data.caltech.edu/records/mzrjq-6wc02.
[6]https://www.kaggle.com/datasets/nltkdata/reuters.
[7]https://www.kaggle.com/datasets/xinleili/nuswide.
[8]http://archive.ics.uci.edu/ml/datasets.

The **Reuters dataset** is a widely adopted benchmark in multi-view learning research. It comprises 18,758 news articles collected by Reuters in 1987, which are categorized into six distinct topics. While the original corpus consists solely of English text, the multi-view variant augments these documents with translations into four additional European languages: French, German, Italian, and Spanish. Consequently, each sample is represented by five distinct views corresponding to these languages. Regarding feature representation, the documents are encoded as Bag-of-Words (BoW) vectors weighted by TF-IDF. The specific feature dimensionality for each view is as follows: 21,531 for English, 24,892 for French, 34,251 for German, 15,506 for Italian, and 11,547 for Spanish.

The **NUS-WIDE dataset**, released by the National University of Singapore, is a large-scale multi-label image dataset comprising 8 distinct categories. It stands as a milestone benchmark in the fields of web image retrieval, multimodal learning, and semantic annotation research. The dataset provides both visual content and associated textual metadata, including user-contributed tags, titles, and descriptions from Flickr, enabling comprehensive cross-modal analysis. In this work, we extract four complementary feature modalities from NUS-WIDE, each capturing distinct aspects of visual and structural information:

- I1 (Cube Color Histogram): A 64-dimensional color histogram computed in a perceptually robust color space. Images are first transformed from the RGB color space to a color space less sensitive to illumination variations, and color distributions are then quantified to obtain global color representations.

- I2 (Color Correlation Map): A 144-dimensional color correlogram that captures both color statistics and their spatial correlations. This representation models not only the occurrence probability of individual colors but also the co-occurrence of identical or different colors at predefined spatial distances.

- I3 (Block-wise Color Moments, BOC): A 225-dimensional block-wise color moments feature that preserves spatial layout information. Each image is uniformly partitioned into a 5×5 grid, and low-order color moments are computed within each block across multiple color channels. The resulting block-level features are concatenated to form the final representation.

- L (Bag-of-Visual-Words): A 500-dimensional bag-of-visual-words representation that treats an image as a document composed of visual words. Local keypoints are first detected and described using local feature descriptors, which are then clustered to construct a visual vocabulary. Each image is subsequently encoded as a histogram over the learned visual words.

The **YouTube dataset** comprises a large collection of short video clips sourced from the YouTube platform, organized around specific events or actions and spanning 10 distinct categories. As a naturally multimodal data source, it provides synchronized visual, auditory, and temporal information. In this study, we extract six complementary feature modalities to capture diverse aspects of the video content:

- I1 (Spatiotemporal Cuboids): A 2,000-dimensional spatiotemporal interest point descriptor that captures local regions exhibiting significant variations across both spatial and temporal dimensions. This representation models local motion patterns and appearance changes using three-dimensional spatiotemporal cuboids, enabling effective characterization of dynamic visual structures.

- I2 (Optical Flow Histogram): A 1,024-dimensional temporally pooled histogram of oriented gradients representation that encodes global shape and edge information. HOG features are extracted frame-wise and aggregated via temporal pooling operations to obtain a compact global descriptor while preserving spatial structural cues.

- T (Frame-level HOG): A 64-dimensional global optical flow histogram that summarizes pixel-level motion statistics. Optical flow fields are computed between consecutive frames, and the distributions of flow magnitudes and orientations are statistically modeled to form a concise representation of overall motion dynamics.

- A1 (Mel-Frequency Features): A 512-dimensional Mel-frequency cepstral coefficient representation that characterizes the spectral properties of audio signals. The audio stream is segmented into frames and processed through windowing, short-time Fourier transform, Mel-scale filtering, and discrete cosine transform, yielding perceptually robust acoustic features.

- A2 (Volume Streams): A 64-dimensional energy-based descriptor that captures variations in audio intensity over time. Frame-level energy statistics are computed using squared amplitude or root mean square measures and aggregated to reflect global loudness dynamics.

- A3 (Spectrogram-based Features): A 647-dimensional spectrogram-based statistical representation that models the joint time–frequency distribution of the audio signal. Spectrograms are generated via short-time Fourier transform and partitioned along both temporal and frequency axes, with energy statistics aggregated within each region to provide a hierarchical description of audio structure.

To accurately simulate complex data distribution scenarios in heterogeneous multimodal federated learning, this study designs three client modality distribution configurations. The "M2" setup refers to each client randomly selecting two modalities from the complete modality set to form its local dataset; the "M1+" setup requires each client to possess at least one modality, while allowing variations in modality composition across different clients; and the "M1" setup restricts each client to holding only a single modality. The number of clients is equal to the number of modalities. In Figures 9, 10, and 11, we visually present the client modality distribution across four datasets. In these visualizations, each circle represents a client, with the circle's area proportional to the sample size of the client's data for the corresponding modality, intuitively illustrating the diversity of modality distribution and the imbalance in data volume under heterogeneous settings.

To evaluate the performance of the algorithm in larger-scale client environments, we scale the number of clients for the four datasets as follows: dataset Caltech101 to 50 clients, dataset Reuters to 100 clients, and datasets NUS-WIDE and YouTube to 20 clients each. The experiments adopt the "M1+" modality configuration, with a Dirichlet distribution (concentration parameter $\alpha = 0.5$) used to partition the data among clients in a non-IID manner. The resulting client data distributions for the scaled scenarios are visualized in Figure 12.

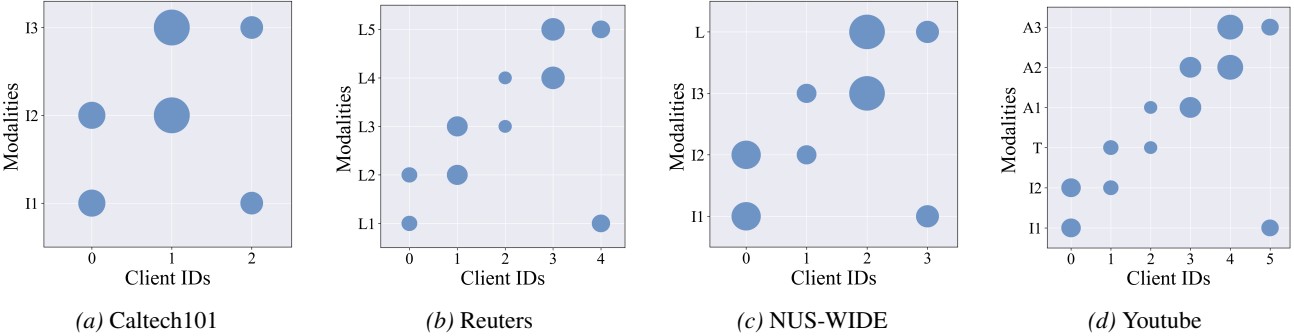

*Figure 9.* The distribution of these datasets is set in the M2 scenario. A larger circle means a larger sample size.

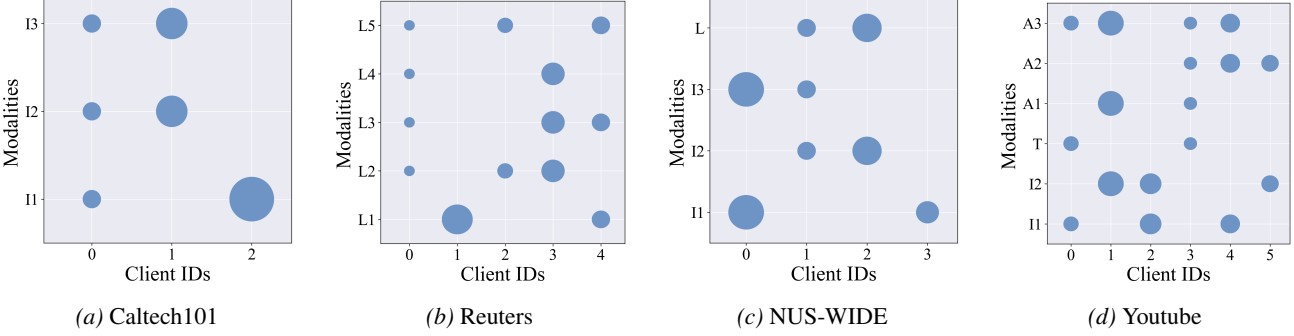

*Figure 10.* The distribution of these datasets is set in the M1+ scenario. A larger circle means a larger sample size.

### C.2. Heterogeneous Model.

After the modality-specific feature preprocessing described in Section C.1, we further assign heterogeneous feature extractors to different modalities in order to obtain modality-specific embeddings. Specifically, the model architecture for each modality

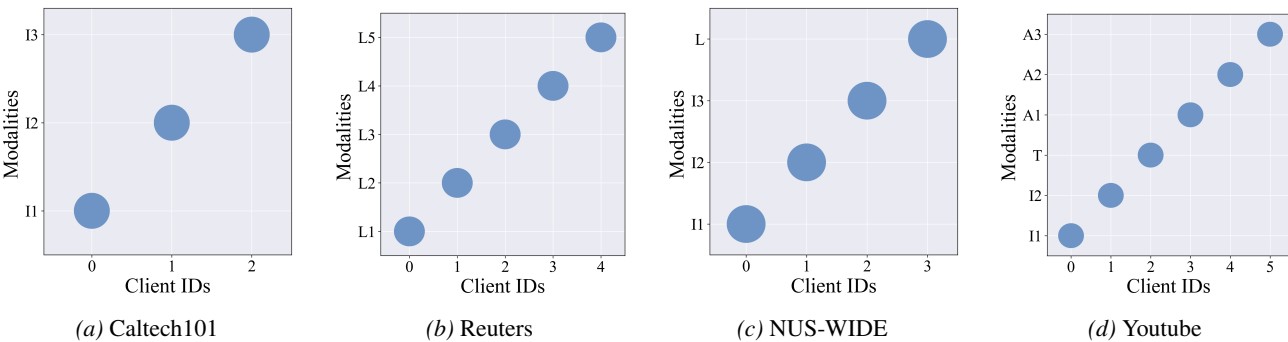

*(a)* Caltech101      *(b)* Reuters      *(c)* NUS-WIDE      *(d)* Youtube

*Figure 11.* The distribution of these datasets is set in the M1 scenario. A larger circle means a larger sample size.

is stochastically sampled from a candidate pool, as listed in Table 4, thereby simulating realistic model heterogeneity in federated learning. To further enforce architectural diversity, the depth (number of repeating blocks) and width (hidden dimensions) of the networks are also randomized.

Table 4 details the sequential structure of these extractors. The symbol "*" denotes the number of repeated modules, which takes integer values in the range of $[2, 5]$ in our experiments. The hidden dimension of each model is set as a multiple of 32 and is kept constant or gradually reduced as the number of modules increases, balancing representational capacity and model complexity. This randomized model assignment strategy enables a controlled yet realistic evaluation of multimodal federated learning under heterogeneous architectures.

*Table 4.* Heterogeneous model architectures for modality-specific feature extraction.

| Model | Sequentially Connected Feature Extractors | Applicable Modality Types |
|---|---|---|
| DNN | [input_dim, *(hidden_dim, Relu), Classifier] | Image, Language, Audio, Time-series |
| CNN1D | [*(Conv1d, Relu, AdaptiveMaxPool1d), *(hidden_dim, Relu), Classifier] | Language, Audio, Time-series |
| CNN2D | [*(Conv2d, Relu), *(hidden_dim, Relu), Classifier] | Image |
| TextCNN | [*(Conv1d, Relu, AdaptiveMaxPool1d), Concat, *(hidden_dim, Relu), Classifier] | Language |
| Resmodel | [input_dim, *ResBlock(hidden_dim,BatchNorm1d, Relu), Classifier] | Audio, Time-series |

### C.3. Baseline Methods

To ensure fair comparison, we re-implemented all baseline methods within a unified HtFLlib framework (Zhang et al., 2025a) and evaluated them under identical experimental settings. Specifically, we include representative prototype-based federated learning methods FedProto (Tan et al., 2022a), FedTGP (Zhang et al., 2024), and FedPall (Zhang et al., 2025c), as well as state-of-the-art multimodal federated learning approaches Harmony (Ouyang et al., 2023), FedMVP (Che et al., 2024), and FedMobile (Liu et al., 2025), with local training (Local) serving as a reference baseline.

Prior to the main experiments, we conducted a preliminary investigation into local multimodal fusion strategies, comparing feature summation ("sum") and feature concatenation ("concat"). Results indicated that feature summation ("sum") yields better performance. Therefore, during local training, for Local, the prototype-based methods, and our proposed MFedPBA, when a client possesses multimodal data, the features extracted by heterogeneous encoders are summed before being sent to the server. In contrast, the multimodal federated learning methods Harmony, FedMVP, and FedMobile follow their original local learning strategies as described in their respective papers.

Furthermore, we note that no publicly available model-heterogeneous multimodal federated learning methods are directly applicable for comparative evaluation. To establish a reasonable heterogeneous experimental environment, we selected state-of-the-art homogeneous multimodal federated learning methods as baselines and adapted their communication mechanisms by modifying the uploaded parameters from full local models to classifier parameters only, thereby accommodating model-heterogeneous federated learning scenarios.

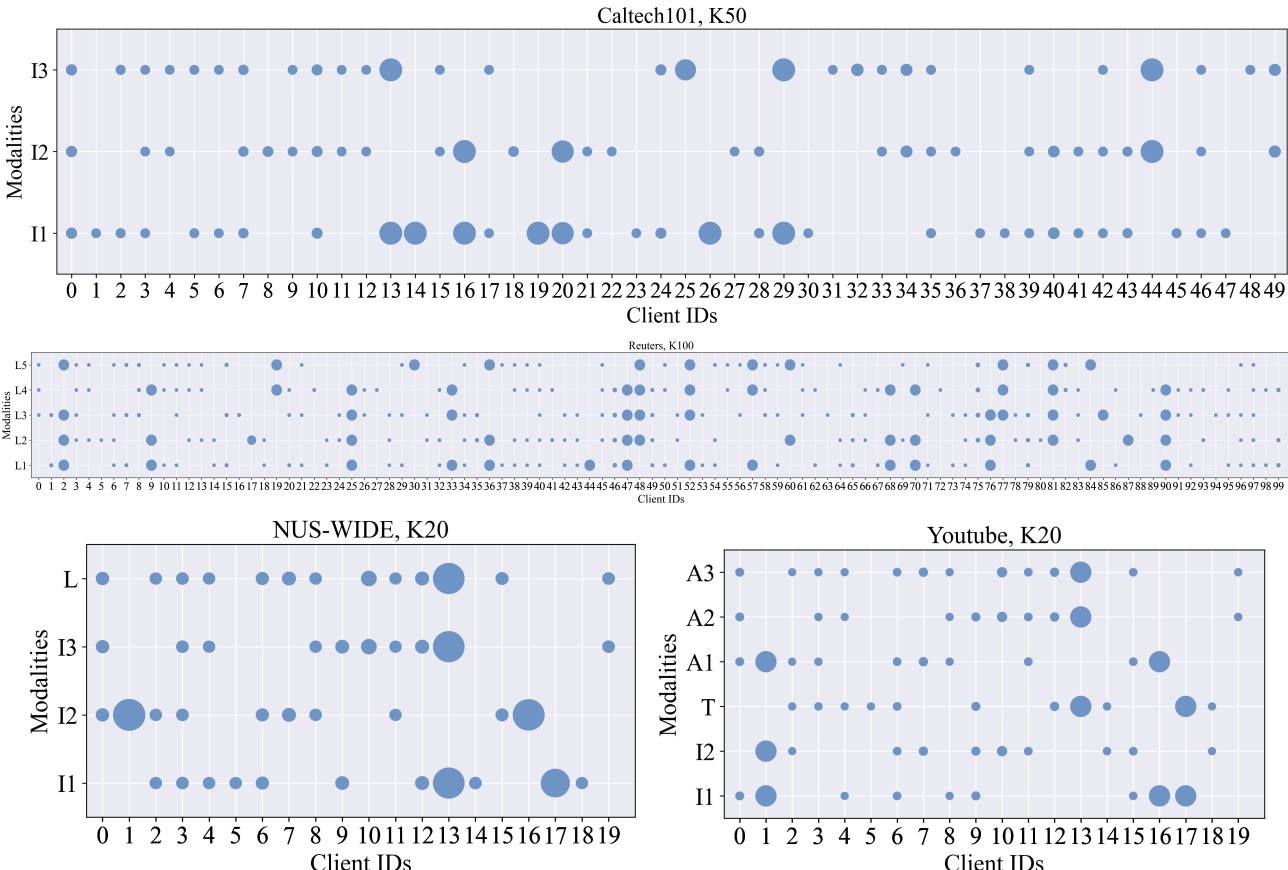

*Figure 12.* In scenarios involving a larger number of clients, the data distribution across these four datasets within the M1+ scenario.

### C.4. Parameter Setting Details

The parameters of our method are detailed in Table 5. We ran these experiments on 2 NVIDIA GeForce RTX 4090 GPUs with AMD Ryzen 9 9950X 16-Core Processor (32 threads) for all methods.

## D. Details of the Experimental Result

In the main text, we report the test accuracy of each method across four multimodal datasets under three data partitioning schemes: "M2", "M1+", and "M1". In the appendix, we provide detailed client-level experimental results and performance curves for each data partitioning scheme.

Regarding the evaluation metrics, the accuracy for an individual client $k$ is defined as the ratio of correctly classified samples ($N_{k,correct}$) to the total number of samples on that client i.e., $acc_k = \frac{N_{k,correct}}{N_k^{test}}$. To account for the influence of data volume, the aggregated accuracy (Avg) reported in the tables is calculated as the arithmetic mean of client accuracies, expressed as: $Avg = \frac{\sum_{k=1}^{K} N_{k,correct}}{\sum_{k=1}^{K} N_k^{test}}$.

The organization of the experimental results corresponding to each setting is as follows: Under the M2 partitioning scheme, the data partition is illustrated in Figure 9. The corresponding test results and performance curves are presented in Table 6 and Figure 13, respectively. Similarly, results for the M1+ scheme are presented in Figure 10, Table 7, and Figure 14, while those for the M1 scheme are provided in Figure 11, Table 8, and Figure 15, respectively.

Furthermore, we extend the evaluation to a larger client pool to assess the scalability of the proposed framework. Based on the dataset size, Dataset Caltech101 is distributed across 50 clients, Dataset Reuters is scaled to 100 clients, and Datasets NUS-WIDE and Youtube are partitioned into 20 clients each, following the M1+ partitioning scheme (as illustrated in Figure

*Table 5.* List of Hyperparameters.

|  | Descriptions | Caltech101 | Reuters | NUS-WIDE | Youtube |
|---|---|---|---|---|---|
|  | Total rounds $T$ | 400 / 500 | 200 | 400 | 400 |
| Server | Server optimizers | SGD | | | |
|  | Server learning rate $\eta_s$ | 0.005~0.01 | | | |
|  | Server training epoch $S$ | 10 | | | |
| Client | Local training epoch $E$ | 2 | | | |
|  | Training batch size $\mathcal{B}$ | 12 | | | |
|  | Local optimizers | SGD | | | |
|  | Weighting parameter $\lambda$ | $\lambda_1 \in \{0.01, 0.01, 0.1\}, \lambda_2 \in \{0.5, 1, 5\}$ | | | |
|  | Local learning rate $\eta_c$ | 0.005~0.01 | 0.001~0.005 | 0.01 | 0.005~0.01 |
|  | Feature embedding dimension $d_D$ | 64 | 24 | 32 | 48 |

12). The models are trained until stable convergence is achieved. The detailed experimental configurations are summarized in the row labeled "K#" in Table 2, and the resulting performance curves are depicted in Figure 16. The experimental results substantiate that MFedPBA consistently maintains superior performance and resilience, even within large-scale federated learning scenarios.

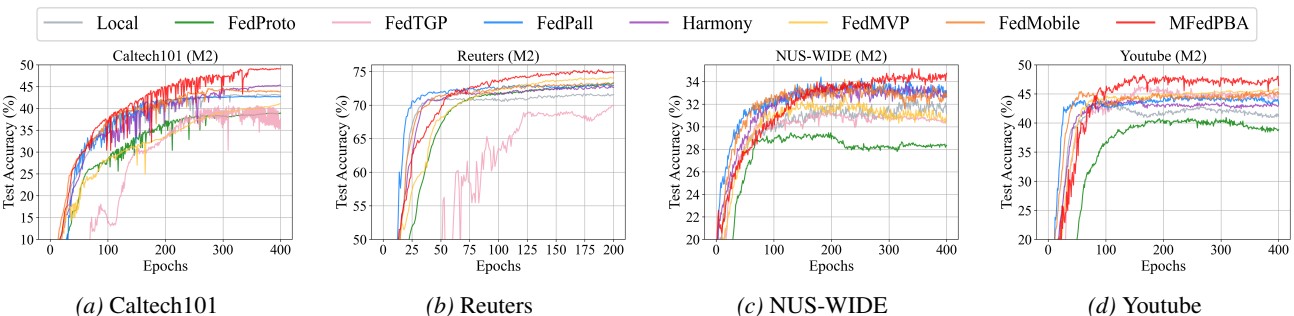

*Figure 13.* The test accuracy and convergence process of each method in M2 scenario.

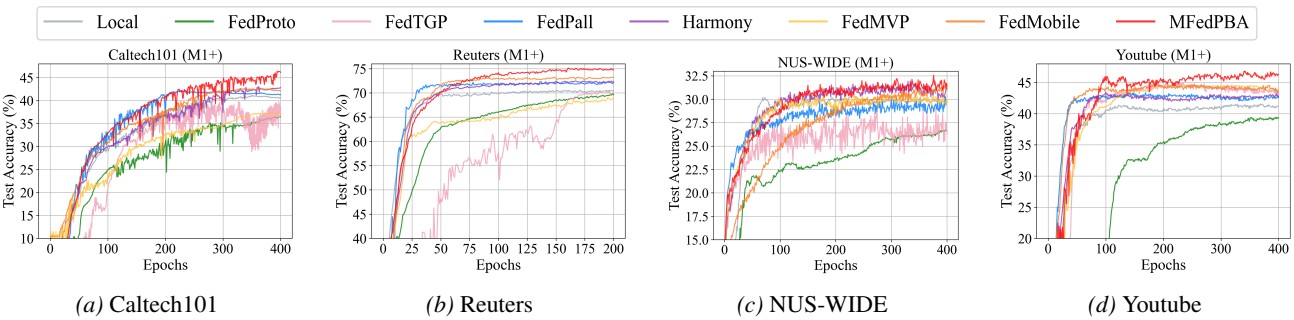

*Figure 14.* The test accuracy and convergence process of each method in M1+ scenario.

*Table 6.* Performance comparison (%) of all compared methods on Caltech101, Reuters, NUS-WIDE, and Youtube using M2 data partitioning, where the number of clients $K$ is equal to the number of modalities $M$. The $k1$ represents the client with ID 1.

| Dataset | | Local | FedProto | FedTGP | FedPall | Harmony | FedMVP | FedMobile | **MFedPBA** |
|---|---|---|---|---|---|---|---|---|---|
| Caltech101 | k1 | $40.48_{\pm0.86}$ | $37.96_{\pm0.75}$ | $39.71_{\pm1.14}$ | $39.81_{\pm0.96}$ | $42.14_{\pm0.17}$ | $35.01_{\pm0.96}$ | $41.89_{\pm1.07}$ | $47.35_{\pm0.19}$ |
| | k2 | $43.28_{\pm0.42}$ | $38.56_{\pm0.21}$ | $39.86_{\pm0.94}$ | $44.18_{\pm0.16}$ | $46.44_{\pm0.15}$ | $41.28_{\pm0.42}$ | $44.72_{\pm0.28}$ | $48.19_{\pm0.18}$ |
| | k3 | $44.24_{\pm0.46}$ | $39.89_{\pm0.15}$ | $43.52_{\pm1.65}$ | $43.3_{\pm0.15}$ | $47.23_{\pm0.23}$ | $44.8_{\pm0.49}$ | $43.52_{\pm0.14}$ | $52.44_{\pm0.27}$ |
| | Avg | $42.65_{\pm1.47}$ | $38.72_{\pm0.42}$ | $40.78_{\pm1.31}$ | $42.29_{\pm0.27}$ | $45.29_{\pm0.38}$ | $40.23_{\pm0.65}$ | $43.51_{\pm0.54}$ | $49.04_{\pm0.21}$ |
| Reuters | k1 | $77.22_{\pm1.58}$ | $82.06_{\pm0.32}$ | $78.63_{\pm0.78}$ | $76.92_{\pm0.62}$ | $76.94_{\pm0.91}$ | $82.53_{\pm1.67}$ | $80.8_{\pm1.07}$ | $82.55_{\pm0.79}$ |
| | k2 | $66.82_{\pm1.42}$ | $64.49_{\pm0.65}$ | $62.23_{\pm1.46}$ | $66.88_{\pm0.61}$ | $68.12_{\pm1.05}$ | $65.62_{\pm1.11}$ | $68.11_{\pm0.38}$ | $67.83_{\pm0.78}$ |
| | k3 | $76.69_{\pm0.98}$ | $79.83_{\pm1.32}$ | $73.33_{\pm1.47}$ | $79.87_{\pm0.23}$ | $79.43_{\pm2.11}$ | $82.44_{\pm0.61}$ | $79.46_{\pm1.48}$ | $79.66_{\pm1.21}$ |
| | k4 | $69.86_{\pm1.09}$ | $73.71_{\pm0.51}$ | $70.78_{\pm0.82}$ | $73.77_{\pm0.39}$ | $69.14_{\pm1.06}$ | $71.52_{\pm1.67}$ | $70.87_{\pm1.17}$ | $75.71_{\pm0.72}$ |
| | k5 | $65.84_{\pm1.05}$ | $71.12_{\pm0.46}$ | $66.76_{\pm1.34}$ | $71.11_{\pm0.33}$ | $71.48_{\pm1.21}$ | $73.78_{\pm0.58}$ | $70.97_{\pm1.66}$ | $73.29_{\pm0.87}$ |
| | Avg | $70.6_{\pm0.28}$ | $73.39_{\pm0.57}$ | $69.73_{\pm0.88}$ | $73.08_{\pm1.04}$ | $72.18_{\pm0.41}$ | $74.07_{\pm1.08}$ | $73.19_{\pm0.26}$ | $75.16_{\pm0.45}$ |
| NUS-WIDE | k1 | $35.28_{\pm0.53}$ | $32.36_{\pm0.59}$ | $35.28_{\pm0.17}$ | $38.97_{\pm1.77}$ | $38.43_{\pm0.95}$ | $37.57_{\pm0.98}$ | $39.26_{\pm1.03}$ | $38.91_{\pm0.78}$ |
| | k2 | $35.76_{\pm1.54}$ | $32.46_{\pm0.66}$ | $34.26_{\pm0.48}$ | $34.72_{\pm0.83}$ | $35.68_{\pm1.37}$ | $32.74_{\pm0.46}$ | $36.47_{\pm1.00}$ | $39.03_{\pm1.00}$ |
| | k3 | $29.52_{\pm0.83}$ | $26.86_{\pm0.12}$ | $28.42_{\pm1.02}$ | $29.9_{\pm0.18}$ | $28.9_{\pm0.84}$ | $30.08_{\pm0.57}$ | $30.16_{\pm0.43}$ | $32.66_{\pm0.42}$ |
| | k4 | $29.31_{\pm0.71}$ | $26.42_{\pm1.47}$ | $27.65_{\pm0.65}$ | $30.48_{\pm1.04}$ | $30.04_{\pm0.22}$ | $26.71_{\pm0.5}$ | $30.5_{\pm0.54}$ | $31.14_{\pm1.53}$ |
| | Avg | $32.18_{\pm0.64}$ | $29.28_{\pm0.46}$ | $31.19_{\pm0.51}$ | $33.38_{\pm0.91}$ | $32.97_{\pm0.28}$ | $31.89_{\pm0.4}$ | $33.86_{\pm0.22}$ | $35.2_{\pm0.46}$ |
| Youtube | k1 | $26.82_{\pm0.33}$ | $31.99_{\pm1.85}$ | $36.97_{\pm1.33}$ | $33.14_{\pm0.88}$ | $31.61_{\pm1.15}$ | $35.44_{\pm0.33}$ | $34.67_{\pm0.33}$ | $39.46_{\pm1.45}$ |
| | k2 | $23.77_{\pm1.12}$ | $22.3_{\pm4.05}$ | $22.55_{\pm0.42}$ | $23.04_{\pm0.42}$ | $21.81_{\pm1.12}$ | $23.53_{\pm0.74}$ | $22.06_{\pm1.27}$ | $21.12_{\pm0.74}$ |
| | k3 | $42.24_{\pm1.49}$ | $35.63_{\pm1.32}$ | $36.49_{\pm1.00}$ | $39.94_{\pm0.5}$ | $43.1_{\pm2.28}$ | $38.22_{\pm0.5}$ | $43.68_{\pm3.03}$ | $46.84_{\pm1.32}$ |
| | k4 | $50.86_{\pm1.57}$ | $51.55_{\pm0.89}$ | $55.5_{\pm3.31}$ | $52.23_{\pm1.81}$ | $55.15_{\pm1.79}$ | $58.08_{\pm1.66}$ | $52.06_{\pm0.89}$ | $55.5_{\pm2.84}$ |
| | k5 | $45.83_{\pm1.74}$ | $44.83_{\pm1.55}$ | $51.01_{\pm2.53}$ | $53.02_{\pm1.88}$ | $50.14_{\pm1.51}$ | $52.3_{\pm0.25}$ | $50.57_{\pm1.99}$ | $54.18_{\pm0.5}$ |
| | k6 | $56.28_{\pm1.5}$ | $53.46_{\pm0.99}$ | $58.44_{\pm2.25}$ | $55.41_{\pm2.46}$ | $53.68_{\pm3.07}$ | $55.84_{\pm1.12}$ | $59.52_{\pm1.35}$ | $60.92_{\pm0.2}$ |
| | Avg | $41.72_{\pm0.66}$ | $41.12_{\pm0.68}$ | $45.16_{\pm1.02}$ | $44.23_{\pm0.3}$ | $43.8_{\pm0.76}$ | $45.53_{\pm0.52}$ | $44.83_{\pm0.53}$ | $47.92_{\pm0.93}$ |

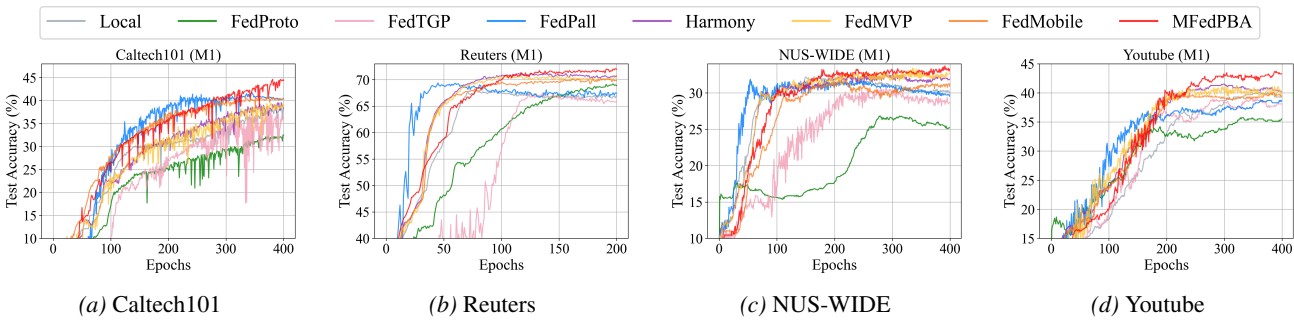

*Figure 15.* The test accuracy and convergence process of each method in M1 scenario.

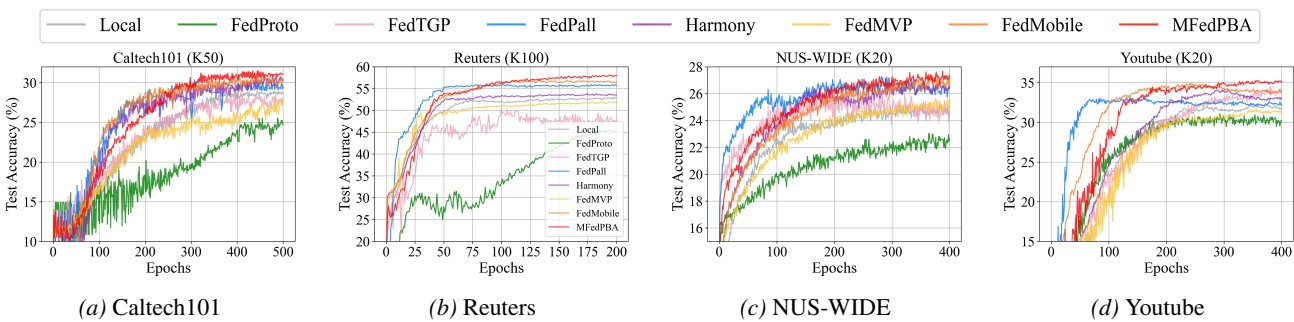

*Figure 16.* The testing accuracy and convergence process of each method in the M1+ scenario, with a large number of clients participating.

*Table 7.* Performance comparison (%) of all compared methods on Caltech101, Reuters, NUS-WIDE, and Youtube using M1+ data partitioning, where the number of clients $K$ is equal to the number of modalities $M$. The $k1$ represents the client with ID 1.

| Dataset | | Local | FedProto | FedTGP | FedPall | Harmony | FedMVP | FedMobile | **MFedPBA** |
|---|---|---|---|---|---|---|---|---|---|
| Caltech101 | k1 | $34.51_{\pm0.67}$ | $30.74_{\pm1.09}$ | $34.7_{\pm1.53}$ | $35.17_{\pm0.79}$ | $39.34_{\pm0.65}$ | $29.58_{\pm1.00}$ | $36.58_{\pm0.61}$ | $41.18_{\pm0.24}$ |
| | k2 | $40.63_{\pm0.87}$ | $35.02_{\pm0.39}$ | $38.57_{\pm0.77}$ | $41.68_{\pm0.83}$ | $41.04_{\pm1.03}$ | $33.69_{\pm0.54}$ | $41.05_{\pm0.58}$ | $45.46_{\pm0.47}$ |
| | k3 | $43.49_{\pm0.58}$ | $40.46_{\pm0.24}$ | $44.59_{\pm2.03}$ | $44.53_{\pm1.22}$ | $45.92_{\pm0.56}$ | $46.3_{\pm1.1}$ | $46.45_{\pm1.57}$ | $50.02_{\pm0.59}$ |
| | Avg | $40.83_{\pm1.12}$ | $36.81_{\pm0.35}$ | $40.71_{\pm1.23}$ | $41.8_{\pm0.77}$ | $43.04_{\pm0.64}$ | $38.91_{\pm1.25}$ | $42.29_{\pm0.6}$ | $46.82_{\pm0.54}$ |
| Reuters | k1 | $72.28_{\pm1.92}$ | $75.00_{\pm2.54}$ | $74.61_{\pm2.58}$ | $74.8_{\pm1.76}$ | $74.71_{\pm0.85}$ | $61.12_{\pm0.88}$ | $75.12_{\pm1.25}$ | $80.82_{\pm1.05}$ |
| | k2 | $82.11_{\pm0.77}$ | $81.2_{\pm0.81}$ | $80.04_{\pm1.56}$ | $81.9_{\pm1.17}$ | $80.76_{\pm0.29}$ | $81.63_{\pm0.23}$ | $83.32_{\pm0.18}$ | $83.72_{\pm0.64}$ |
| | k3 | $60.57_{\pm0.85}$ | $53.4_{\pm1.25}$ | $57.83_{\pm1.87}$ | $60.12_{\pm1.18}$ | $61.7_{\pm0.67}$ | $60.05_{\pm0.34}$ | $61.31_{\pm0.14}$ | $62.21_{\pm0.7}$ |
| | k4 | $65.57_{\pm0.53}$ | $58.42_{\pm1.32}$ | $61.35_{\pm1.94}$ | $63.04_{\pm2.47}$ | $67.64_{\pm0.25}$ | $64.27_{\pm0.67}$ | $67.06_{\pm0.45}$ | $68.34_{\pm0.46}$ |
| | k5 | $73.95_{\pm2.65}$ | $77.74_{\pm1.01}$ | $77.39_{\pm1.57}$ | $78.4_{\pm1.14}$ | $77.61_{\pm0.46}$ | $77.26_{\pm0.27}$ | $78.95_{\pm1.55}$ | $81.82_{\pm0.21}$ |
| | Avg | $70.85_{\pm0.59}$ | $68.54_{\pm1.46}$ | $69.8_{\pm0.59}$ | $71.09_{\pm1.56}$ | $72.28_{\pm0.47}$ | $68.09_{\pm0.82}$ | $72.79_{\pm0.44}$ | $75.01_{\pm0.22}$ |
| NUS-WIDE | k1 | $28.63_{\pm0.75}$ | $30.21_{\pm0.43}$ | $27.48_{\pm1.11}$ | $30.77_{\pm1.41}$ | $29.82_{\pm0.65}$ | $28.16_{\pm0.9}$ | $32.82_{\pm0.53}$ | $33.02_{\pm1.31}$ |
| | k2 | $28.17_{\pm0.24}$ | $26.2_{\pm0.68}$ | $25.67_{\pm1.39}$ | $29.35_{\pm0.42}$ | $31.83_{\pm1.24}$ | $29.51_{\pm0.53}$ | $29.77_{\pm0.55}$ | $30.9_{\pm0.85}$ |
| | k3 | $28.7_{\pm1.26}$ | $25.94_{\pm0.81}$ | $26.37_{\pm2.01}$ | $31.01_{\pm0.81}$ | $32.42_{\pm0.59}$ | $30.92_{\pm0.51}$ | $29.61_{\pm0.73}$ | $31.93_{\pm0.33}$ |
| | k4 | $34.32_{\pm2.79}$ | $28.75_{\pm0.57}$ | $28.52_{\pm1.91}$ | $35.56_{\pm0.86}$ | $34.21_{\pm1.97}$ | $32.96_{\pm1.2}$ | $31.35_{\pm3.97}$ | $33.71_{\pm2.53}$ |
| | Avg | $28.72_{\pm0.71}$ | $27.79_{\pm0.55}$ | $27.13_{\pm0.18}$ | $30.99_{\pm0.63}$ | $31.56_{\pm0.61}$ | $29.82_{\pm0.59}$ | $30.95_{\pm0.57}$ | $32.15_{\pm0.36}$ |
| Youtube | k1 | $33.5_{\pm1.02}$ | $33.33_{\pm0.85}$ | $33.17_{\pm2.04}$ | $35.62_{\pm1.02}$ | $35.78_{\pm1.7}$ | $35.46_{\pm0.57}$ | $40.23_{\pm2.19}$ | $40.52_{\pm1.02}$ |
| | k2 | $57.18_{\pm2.46}$ | $52.87_{\pm0.5}$ | $58.05_{\pm2.24}$ | $56.7_{\pm1.63}$ | $57.95_{\pm1.68}$ | $59.58_{\pm0.66}$ | $58.14_{\pm0.66}$ | $60.25_{\pm0.44}$ |
| | k3 | $37.5_{\pm1.38}$ | $30.56_{\pm4.35}$ | $37.5_{\pm1.56}$ | $36.63_{\pm1.97}$ | $37.15_{\pm1.31}$ | $34.55_{\pm0.3}$ | $37.15_{\pm0.6}$ | $36.81_{\pm1.06}$ |
| | k4 | $40.37_{\pm1.79}$ | $37.93_{\pm4.48}$ | $44.54_{\pm1.00}$ | $42.82_{\pm1.79}$ | $43.39_{\pm3.06}$ | $42.24_{\pm0.86}$ | $44.4_{\pm0.75}$ | $45.4_{\pm0.25}$ |
| | k5 | $44.32_{\pm1.97}$ | $40.87_{\pm4.2}$ | $49.68_{\pm0.96}$ | $47.51_{\pm1.38}$ | $47.13_{\pm0.66}$ | $49.17_{\pm0.96}$ | $46.49_{\pm1.73}$ | $50.45_{\pm1.33}$ |
| | k6 | $22.08_{\pm1.12}$ | $22.51_{\pm5.32}$ | $29.06_{\pm2.15}$ | $22.73_{\pm2.83}$ | $22.73_{\pm2.34}$ | $27.92_{\pm2.34}$ | $27.27_{\pm2.34}$ | $29.22_{\pm2.34}$ |
| | Avg | $41.89_{\pm0.76}$ | $38.82_{\pm1.12}$ | $44.55_{\pm0.68}$ | $43.04_{\pm1.09}$ | $43.47_{\pm0.91}$ | $44.24_{\pm0.36}$ | $44.28_{\pm0.64}$ | $47.21_{\pm0.59}$ |

*Table 8.* Performance comparison (%) of all compared methods on Caltech101, Reuters, NUS-WIDE, and Youtube using M1 data partitioning, where the number of clients $K$ is equal to the number of modalities $M$. The X1 represents a client that has the X1 modal enabled.

| Dataset | | Local | FedProto | FedTGP | FedPall | Harmony | FedMVP | FedMobile | **MFedPBA** |
|---|---|---|---|---|---|---|---|---|---|
| Caltech101 | I1 | $39.47_{\pm0.13}$ | $38.27_{\pm1.4}$ | $41.16_{\pm0.67}$ | $42_{\pm0.71}$ | $39.84_{\pm0.62}$ | $38.62_{\pm0.5}$ | $41.87_{\pm0.13}$ | $48.73_{\pm0.9}$ |
| | I2 | $36.89_{\pm0.28}$ | $25.56_{\pm2.04}$ | $31.47_{\pm1.73}$ | $37.02_{\pm0.56}$ | $39.78_{\pm1.3}$ | $38.09_{\pm1.16}$ | $38.17_{\pm0.73}$ | $40.78_{\pm0.37}$ |
| | I3 | $39.07_{\pm1.22}$ | $30.89_{\pm1.34}$ | $35.91_{\pm0.66}$ | $43.29_{\pm0.2}$ | $40.69_{\pm0.83}$ | $40.04_{\pm1.65}$ | $40.22_{\pm0.28}$ | $41.4_{\pm1.05}$ |
| | Avg | $38.47_{\pm0.45}$ | $31.57_{\pm1.23}$ | $36.18_{\pm0.72}$ | $40.77_{\pm0.36}$ | $40.1_{\pm0.69}$ | $38.92_{\pm0.62}$ | $40.09_{\pm0.13}$ | $43.64_{\pm0.11}$ |
| Reuters | L1 | $64.47_{\pm1.07}$ | $58.87_{\pm0.61}$ | $60.39_{\pm0.93}$ | $65.85_{\pm0.55}$ | $64.39_{\pm0.55}$ | $63.85_{\pm1.59}$ | $64.93_{\pm0.43}$ | $67.2_{\pm1.33}$ |
| | L2 | $69.81_{\pm1.4}$ | $79.52_{\pm0.67}$ | $77.58_{\pm0.88}$ | $74.31_{\pm2.03}$ | $76.44_{\pm0.18}$ | $76.41_{\pm0.25}$ | $75.37_{\pm0.38}$ | $77.87_{\pm0.85}$ |
| | L3 | $63.15_{\pm0.96}$ | $66.7_{\pm0.71}$ | $48.43_{\pm0.99}$ | $54.55_{\pm3.67}$ | $67.38_{\pm0.43}$ | $68.61_{\pm1.21}$ | $67.31_{\pm0.44}$ | $68.26_{\pm0.33}$ |
| | L4 | $72.39_{\pm1.59}$ | $78.29_{\pm0.16}$ | $76.61_{\pm1.64}$ | $73.24_{\pm1.29}$ | $76.19_{\pm0.12}$ | $75.01_{\pm1.12}$ | $75.2_{\pm0.62}$ | $79.83_{\pm1.65}$ |
| | L5 | $65.44_{\pm1.55}$ | $60.38_{\pm0.65}$ | $66.04_{\pm0.77}$ | $66.84_{\pm0.98}$ | $66.42_{\pm0.7}$ | $67.04_{\pm0.13}$ | $66.81_{\pm0.33}$ | $67.31_{\pm0.33}$ |
| | Avg | $67.05_{\pm1.04}$ | $68.75_{\pm0.63}$ | $65.81_{\pm0.33}$ | $66.96_{\pm0.22}$ | $70.16_{\pm1.14}$ | $70.18_{\pm0.57}$ | $69.92_{\pm0.95}$ | $72.09_{\pm0.79}$ |
| NUS-WIDE | I1 | $34.5_{\pm0.85}$ | $23.15_{\pm1.55}$ | $35.04_{\pm0.18}$ | $32.27_{\pm3.24}$ | $34.4_{\pm1.12}$ | $33.33_{\pm1.21}$ | $33.65_{\pm1.34}$ | $34.29_{\pm0.8}$ |
| | I2 | $36.85_{\pm0.49}$ | $32.32_{\pm1.34}$ | $34.08_{\pm3.02}$ | $36.00_{\pm0.49}$ | $38.76_{\pm1.58}$ | $39.4_{\pm0.37}$ | $36.53_{\pm1.76}$ | $40.79_{\pm0.85}$ |
| | I3 | $32.37_{\pm0.8}$ | $22.17_{\pm1.63}$ | $30.35_{\pm0.86}$ | $35.46_{\pm1.15}$ | $33.01_{\pm1.29}$ | $34.61_{\pm1.21}$ | $33.55_{\pm1.66}$ | $35.46_{\pm0.68}$ |
| | L | $20.87_{\pm0.49}$ | $22.19_{\pm3.02}$ | $18.74_{\pm0.82}$ | $20.34_{\pm0.74}$ | $22.58_{\pm1.61}$ | $23.43_{\pm0.78}$ | $21.09_{\pm0.32}$ | $22.47_{\pm0.49}$ |
| | Avg | $31.15_{\pm0.14}$ | $24.96_{\pm0.52}$ | $29.55_{\pm0.73}$ | $31.02_{\pm0.96}$ | $32.19_{\pm0.35}$ | $32.7_{\pm0.51}$ | $31.2_{\pm0.36}$ | $33.25_{\pm0.24}$ |
| Youtube | I1 | $34.13_{\pm0.69}$ | $39.29_{\pm2.06}$ | $32.94_{\pm2.75}$ | $34.13_{\pm2.48}$ | $33.33_{\pm2.06}$ | $34.34_{\pm1.74}$ | $37.3_{\pm0.69}$ | $44.44_{\pm1.37}$ |
| | I2 | $34.13_{\pm1.37}$ | $26.9_{\pm2.89}$ | $27.78_{\pm1.37}$ | $33.33_{\pm1.19}$ | $41.27_{\pm3.64}$ | $38.89_{\pm1.82}$ | $34.52_{\pm3.15}$ | $30.56_{\pm1.37}$ |
| | T | $22.22_{\pm1.82}$ | $18.2_{\pm0.59}$ | $21.83_{\pm0.69}$ | $21.43_{\pm3.15}$ | $22.62_{\pm2.06}$ | $21.37_{\pm1.29}$ | $20.24_{\pm2.38}$ | $28.56_{\pm1.05}$ |
| | A1 | $67.06_{\pm1.37}$ | $60.71_{\pm1.19}$ | $57.54_{\pm3.44}$ | $57.14_{\pm2.38}$ | $64.68_{\pm2.48}$ | $60.32_{\pm1.81}$ | $56.4_{\pm0.77}$ | $71.37_{\pm1.2}$ |
| | A2 | $32.94_{\pm3.44}$ | $24.81_{\pm1.73}$ | $32.94_{\pm2.48}$ | $35.71_{\pm2.06}$ | $34.13_{\pm1.82}$ | $33.33_{\pm1.19}$ | $34.87_{\pm1.74}$ | $30.75_{\pm2.22}$ |
| | A3 | $49.6_{\pm1.37}$ | $49.61_{\pm1.39}$ | $58.73_{\pm0.69}$ | $53.17_{\pm2.48}$ | $50.4_{\pm0.69}$ | $51.98_{\pm2.48}$ | $53.57_{\pm3.15}$ | $61.44_{\pm0.64}$ |
| | Avg | $40.01_{\pm0.41}$ | $36.59_{\pm0.39}$ | $38.62_{\pm0.64}$ | $39.15_{\pm1.5}$ | $41.07_{\pm0.42}$ | $40.04_{\pm0.46}$ | $39.48_{\pm0.99}$ | $44.52_{\pm0.54}$ |

