# OpenReview forum: "Prototype-guided Bilateral Alignment Multimodal Federated Learning"
_ICML.cc/2026/Conference — ICML 2026 spotlight_

### Official Review · Reviewer_cLqt · 2026-03-07

**Soundness:** 4
**Presentation:** 3
**Significance:** 4
**Originality:** 4
**Overall Recommendation:** 5
**Confidence:** 5

**Summary:**

This paper innovatively proposes the MFedPBA framework, which addresses the heterogeneity of client hardware and uneven distribution of modal data in the real world through a two-sided alignment mechanism. At the feature level, a self-encoding alignment framework is introduced on the server side. Contrastive learning is used to enhance intra-modal semantic consistency, and Gromov-Wasserstein (GW) distance is employed to align the geometric structure between the heterogeneous feature space and the global space, thereby achieving effective unification of the feature space. Leveraging the inherent property of logit prototypes existing in a shared semantic space among heterogeneous models, an entropy-based weighted aggregation strategy is used to construct highly discriminative global logit prototypes, achieving decision alignment.

**Compliance With Llm Reviewing Policy:**

Affirmed.

**Final Justification:**

The authors have addressed my concerns. I will maintain my original score.

**Key Questions For Authors:**

please refer to the weaknesses

**Limitations:**

yes

**Strengths And Weaknesses:**

Strengths:

1. Extensive Baseline Support: Fair comparisons were conducted with various baselines, including prototype federated learning and advanced multimodal federated learning, with results consistently showing MFedPBA to significantly outperform the method.

2. Theoretical Support: A theoretical proof of convergence under non-convex settings is provided, increasing the reliability of the method.

3. Extensive Experimentation: Ablation studies, computational/communication overhead analysis, and hyperparameter sensitivity tests were performed, comprehensively evaluating all components of the method and practical deployment considerations.

Weaknesses:

1: Method Explanation: The authors need to explain the optimization process of the GW distance, especially how it uses the GW distance feature space for alignment. This cross-dimensional vector alignment is one of the important highlights of the paper, and the authors should provide a clearer and more detailed explanation.

2: Experimental Explanation: The authors need to explain why the features learned by the local method and their method in Figure 4 show such a significant difference. This will help readers better understand the contribution of their method.

---

> ### Author Rebuttal · Authors · 2026-03-28
>
> Thank you for your question. I will reply point by point
>
> > W1: Regarding GW Distance
>
> We appreciate your insightful comment. As you noted, effective alignment across varying feature dimensions is a core contribution of this work. We provided a brief overview of the Gromov-Wasserstein (GW) Distance in Section 3.3, though space constraints limited the detail.
>
> We employ GW distance because, in our heterogeneous setting, clients use distinct feature extractors due to hardware or modality differences. This results in inconsistent feature dimensions that differ from the final Logit dimensions, rendering traditional alignment methods (e.g., Euclidean distance or Cosine similarity) inapplicable. In contrast, GW distance bypasses direct point-to-point comparison by comparing internal "geometric topologies." We consider two spaces structurally aligned as long as their relative category distance distributions are consistent.
>
> In Section 4.1.2, we detail this alignment and optimization process, which consists of the following steps:
>
> **Step 1: Constructing Intra-space Relation Matrices:**
>
> To circumvent the inconsistency in feature vector dimensions, we shift the alignment focus to the relative distances between categories. Regardless of the absolute feature dimension, the total number of global categories, $C$, remains constant.
>
> On the server side, we compute the pairwise squared Euclidean distances (i.e., $C\_{i,j} = ||x\_i - x\_j||\_2^2$ ) between category prototypes within both the Global Logit Prototype Space $I_{G,c}$ and the Specific Modality Feature Prototype Space $P_{G,c}^m$. This yields two Cost Matrices: $C_{I\_G} \in \mathbb{R}^{d\_C \times d\_C}$ and $C\_{P\_G^m} \in \mathbb{R}^{d\_C \times d\_C}$.
>
> **Step 2: Formulating the GW Distance Objective:**
>
> After mapping the disparate spaces into relation matrices of uniform size, we seek an Optimal Transport Plan $T$ to minimize the relational distortion between these two topologies. Consequently, we utilize Equation 11 to align with the Global Logit space, which is rich in global semantics and inherently free of dimensional discrepancies.
>
> **Step 3: Entropic Regularization and Sinkhorn Algorithm**
>
> The original GW optimization is a highly complex non-convex quadratic assignment problem. Directly solving it is computationally expensive and not suitable for gradient-based learning.
>
> To address this issue, we introduce an entropic regularization term into the objective: $\varepsilon H(T) = -\varepsilon \sum_{i,j} T_{ij} \log T\_{ij}$. This regularization smooths the otherwise discrete transport problem, making the GW computation fully differentiable. Consequently, the optimization can be efficiently solved using the **Sinkhorn algorithm**, which enables scalable and stable computation.
>
> **Step 4: Gradient Backpropagation and Network Update**
>
> Upon obtaining the optimal solution T via Sinkhorn iterations, we compute the structural alignment loss $\mathcal{L}\_{align}$ . Since every tensor operation within the Sinkhorn iterations preserves the computational graph, we can successfully differentiate $\mathcal{L}\_{align}$ and backpropagate gradients to the server-side specific modality feature encoder $\Phi\_m(\cdot)$ and the shared decoder $\psi(\cdot)$.
>
> > W2: About Visualization Results
>
> Thank you for your question.
> Regarding the visualization on the left side of Figure 4 (Local), even samples belonging to the same category exhibit a fragmented, strip-like dispersion where different modalities (indicated by colors) are distant from one another. This phenomenon stems primarily from two factors:
>
> - In our heterogeneous setting, different clients employ completely distinct architectures for feature extraction across modalities. When clients perform isolated training relying solely on limited, imbalanced local data, these heterogeneous networks naturally map inputs to drastically different embedding spaces.
>
> - Local models lack visibility into the data distributions of other clients, making them prone to overfitting local biased data. This results in features that lack generalization capability and fail to form compact class clusters.
>
> Our method demonstrates superior clustering performance because the proposed bilateral alignment framework exerts a strong global constraint during local model updates.

---

> > ### Author Rebuttal · Reviewer_cLqt · 2026-04-01
> >
> > The author's clear response to the GW calculation process has helped me better understand the calculation process of the paper. I am willing to support the acceptance of this paper.

---

> > > ### Author Response · Authors · 2026-04-03
> > >
> > > We sincerely thank you for your professional review and active support. We are deeply grateful for the great amount of effort you have put into this project and the patient guidance you have provided during the revision process.

---

### Official Review · Reviewer_GhuD · 2026-03-11

**Soundness:** 3
**Presentation:** 3
**Significance:** 4
**Originality:** 4
**Overall Recommendation:** 4
**Confidence:** 4

**Summary:**

The authors argue that existing MFL methods are typically based on idealized assumptions of model homogeneity and modal balance, making it difficult to handle real-world scenarios with heterogeneous client hardware and uneven distribution of modal data. Especially when clients use different encoders, directly aggregating feature prototypes can lead to representation misalignment and knowledge contamination, potentially causing the global model to perform worse than a baseline using only local data. Therefore, the authors employ a two-sided alignment approach to help the model learn better performance.

**Compliance With Llm Reviewing Policy:**

Affirmed.

**Key Questions For Authors:**

See weaknesses.

**Limitations:**

Yes.

**Strengths And Weaknesses:**

Strengths:

1. The paper clearly points out and quantifies the advantages of the method in terms of communication overhead. Compared with methods that require transmitting complete model parameters, MFedPBA only transmits feature prototypes and logistic prototypes, which is crucial for bandwidth-constrained federated learning scenarios. Furthermore, the appendix provides a complete analysis of non-convex convergence, increasing its rigor and credibility as academic research.

2. The framework design consciously offloads the main computationally intensive tasks to the resource-rich server, while the client only needs to perform local training and simple prototype alignment loss calculation. This design aligns with the typical assumption of "strong server, weak client" in federated learning, offering better feasibility for engineering deployment.


Weaknesses:

1. The authors did not discuss extreme client scenarios. The core of their method is to compensate for missing modal information on the client by aggregating global prototypes. However, if a client has very little data, its computed local prototype is inherently unreliable. Even if the server provides a high-quality global prototype, the client may still be unable to effectively distill global knowledge into its own model due to insufficient local data.

2. Although the authors provide detailed experimental information in the appendix, they did not test whether local epoch training affected the experimental results.

3. While the paper demonstrates robustness within certain ranges, finding the optimal parameter combination for a specific dataset in real-world deployment may require extensive tuning, increasing the difficulty of application.

4. Although the experiments cover multiple modalities, they are primarily simulations conducted on standard image, text, and audio datasets, and employ a relatively simple fusion method. Could the authors further discuss the benefits of different fusion methods?

5. Minor issue: Eq. (50) has a slight subscript problem.

---

> ### Author Rebuttal · Authors · 2026-03-29
>
> Thank you for the reviewer's questions and support. Next, we will provide a detailed response to the reviewer's questions.
>
> > W1: Discussion on Extreme Clients
>
> We appreciate your concern regarding the potential failure of knowledge distillation due to unreliable local prototypes in extreme scenarios. In fact, we have explicitly considered and addressed this specific client setting in our paper.
>
> In **Section 6.1 (Experimental Setup)**, we designed the "M1" data partition setting, where each client possesses only a single, absolute modality (i.e., completely lacking all other complementary modalities). In the face of such extreme unreliability, traditional federated learning methods would indeed suffer from model collapse or distillation failure due to severe bias. However, MFedPBA maintains superior performance, thanks to our introduced GW distance-based topological alignment mechanism.
>
> When client data is scarce and unimodal, the computed absolute feature vectors are highly unreliable and significantly biased. The effectiveness of GW distance lies in extracting the "relative distance proportions" between local categories rather than relying on absolute positions. As demonstrated in the M1 scenario results in Table 2, MFedPBA achieves significantly better results than all baseline methods.
>
>
>
> > W2: Discussion on Local Epoch
>
> We appreciate the reviewer's question. We systematically evaluated the performance of different methods on the YouTube dataset M1+ setting across varying numbers of local epochs.
>
> | ACC(%)    | E=1   | E=2   | E=3   | E=4   | E=5   |
> | :-------- | :---- | :---- | :---- | :---- | :---- |
> | Local     | 41.12 | 41.89 | 42.42 | 42.46 | 41.56 |
> | FedProto  | 38.53 | 38.82 | 38.85 | 37.42 | 37.12 |
> | FedTGP    | 43.87 | 44.55 | 45.25 | 45.05 | 44.75 |
> | FedPall   | 43.54 | 43.04 | 44.86 | 45.06 | 44.23 |
> | Harmony   | 43.04 | 43.47 | 44.35 | 44.65 | 43.95 |
> | FedMVP    | 43.65 | 44.24 | 45.33 | 45.54 | 44.65 |
> | FedMobile | 44.24 | 44.28 | 44.86 | 45.07 | 44.86 |
> | MFedPBA   | 46.53 | 47.21 | 47.84 | 48.58 | 48.02 |
>
> The experimental results indicate that our method exhibits a consistent performance trend with all other baseline methods. When the number of local epochs increases moderately from a small value, models can more fully exploit local data for feature learning, leading to improved accuracy across all methods. Conversely, when the number of local epochs is excessive, local models severely overfit the biased local data distributions, resulting in significant client drift. Despite following this same trend, MFedPBA consistently and significantly outperforms all baseline methods, regardless of the setting for local training epochs.
>
>
>
> > W3: Discussion on Parameter
>
> We appreciate the reviewer's question. We understand that hyperparameter tuning is a common challenge in the field. However, our method does not rely on precise, delicate tuning. As demonstrated in **Section 6.2 (Figure 7, Hyperparameter experiments)**, MFedPBA is robust and does not function effectively only at a single, extremely specific optimal point.
>
> In our sensitivity analysis, the model's accuracy fluctuates very smoothly even when the weighting parameters λ1 and λ2, which control feature alignment and Logit alignment, vary across a very broad range. Furthermore, within this wide interval, any combination of parameters yields performance that is stable and significantly superior to all baseline methods. This implies that during practical deployment, engineers do not need to perform an exhaustive grid search, provided the parameters are not set to extreme values.
>
>
>
> > W4: Fusion Form
>
> We appreciate the reviewer's discussion regarding fusion forms. We fully agree that advanced, adaptive multimodal fusion mechanisms, such as attention mechanisms, could further enhance model performance.
>
> First, we wish to clarify that the primary focus and core contribution of this work lie not in exploring or innovating local client-side multimodal fusion architectures, but rather in addressing cross-node collaboration challenges within the macroscopic federated learning framework. Specifically, we aim to overcome two major systemic barriers in real-world scenarios: "model architecture heterogeneity" and "severe modality imbalance."
>
> We intentionally employed the most fundamental fusion strategy on local clients primarily for experimental considerations regarding controlled variables and fair comparison. Introducing more complex attention or gating fusion mechanisms locally could obscure the specific contributions of the federated knowledge transfer framework itself.
>
>
>
> > W5: Symbol Problem
>
> Thank you for your careful discovery. We will make revisions accordingly.

---

> > ### Author Rebuttal · Reviewer_GhuD · 2026-04-02
> >
> > After reading the author's experiment and response, the author's reply has resolved most of my questions. Therefore, I support this paper. I also suggest the author could pursue better innovations in client‑side fusion methods in the future. I maintain my score unchanged.

---

> > > ### Author Response · Authors · 2026-04-03
> > >
> > > We would like to express our gratitude once again for your insightful comments and sincere feedback. We are committed to conducting more innovative research in the future.

---

### Official Review · Reviewer_WGff · 2026-03-12

**Soundness:** 3
**Presentation:** 3
**Significance:** 4
**Originality:** 3
**Overall Recommendation:** 4
**Confidence:** 4

**Summary:**

The authors focus on solving an important and timely problem: feature prototype misalignment caused by federated learning (FL) systems under two realistic constraints: heterogeneous model architectures (different modality-specific encoders) and imbalanced modality distributions (clients observe different modality subsets). This paper proposes MFedPBA, equiped with a prototype-guided bilateral alignment framework, designed to address this problem.

The paper provides a nonconvex convergence analysis and extensive experiments across 4 multimodal benchmarks. Empirical results demonstrate consistent improvements over strong baselines.

**Compliance With Llm Reviewing Policy:**

Affirmed.

**Final Justification:**

The authors addressed my concerns. I have raised my score from negative to positive.

**Key Questions For Authors:**

Key Questions For Authors
1. I know that due to the page limit, the authors could not include too much information in the main body. However, to make the work approachable to a broad audience, I suggest the authors should add a new paragraph to the Introduction where appropriate to provide some more context. The new parapgraph includes the definition of what a feature prototype and a logit prototype represent, briefly describe how they are generated (by encoders), what prototype-based FL is, how it works briefly.
2. The GW-based alignment relies on entropy-regularized optimal transport solved via the Sinkhorn algorithm. However, the manuscript does not clearly specify the number of Sinkhorn iterations or the entropy regularization coefficient. Since the authors chose not to disclose their codes during review period, I think they should clarify their practical configuration for reproducibility. For example, what values of Ts and epsilon are used in the paper? Do they suggest default values for them?
3. the Related Work section needs more detailed information. Firstly, when the authors discuss specific existing methods, please include their name with corresponding citations. I do not see some baselines as FedProto, FedTGP, FedPall, etc. here. Secondly, please also discuss their methodology and weaknesses under joint model and modality heterogeneity to highlight the novelty of MFedPBA.
4. Entropy is sensitive to logit magnitude. The entropy-based weighting assumes that logits are directly comparable across clients. However, logit magnitudes can vary substantially under federated heterogeneity, so entropy-based weighting may bias aggregation. Did the authors consider normalizing logits (e.g., temperature scaling or norm normalization) before computing entropy?
Some minor weaknesses:
5. Figure 1b shows an inconsistency. Feature space of Client B: blue feature prototype should be represented as a blue star. $M_k$ in Figure 2 misses the language modality.
6. Since the authors abbreviate multimodal federated learning with MFL, please use MFL throughout the work for consistency. The same for federated learning with FL.
The manuscript involves two distinct types of prototypes: (i) feature prototypes and (ii) logit prototypes. For clarity and consistency, I suggest consistently using these specific terms throughout the paper. In several places, the general terms “class prototypes”, “embedding prototypes”, “global prototype” are used without explicitly distinguishing between feature-level and logit-level prototypes, which may cause ambiguity for readers.

**Limitations:**

yes

**Strengths And Weaknesses:**

Strengths

The work reflected huge efforts from the authors’ side to complete this work. The below is main strengths I could see from the paper:

+ Well-written and explicit motivation. This work presents the problem context clearly and logically builds the case for why naive prototype aggregation fails under heterogeneous encoders. The discussion around representation misalignment and knowledge contamination is intuitive and convincing.

+ Modular and structured design. I can see the logic here: the framework is thoughtfully decomposed into feature level alignment (addressing geometry mismatch) and decision level alignment (leveraging class aligned logits) is intuitive and technically meaningful.. Each module targets a specific weakness of prior approaches.

+ Comprehensive experiments. The evaluation covers multiple datasets and modality partition regimes (M2, M1+, M1), includes scalability experiments with larger client counts, ablations on server side components, feature visualization, hyperparameter sensitivity, and runtime measurements.

Weaknesses

1. The context and preliminaries could be clearer for a broader audience.
2. Unclear Sinkhorn configuration.
3. Unclear existing competitors.
4. Logit prototype comparability.
Please read my below feedback for more details

---

> ### Author Rebuttal · Authors · 2026-03-29
>
> Thank you for the reviewer's review and questions. We will now respond to your questions one by one.
>
> > W1: Prototype Explanation
>
> We sincerely appreciate the reviewer's valuable suggestion aimed at broadening its audience. First, we agree with your view that an intuitive explanation of the definition of prototypes should be added to the Introduction. Second, we wish to clarify that the complete operating mechanism of "prototype-based FL," along with the specific mathematical definitions and generation methods for "feature prototypes" and "Logit prototypes," has already been rigorously formalized in **Section 3.2 (Prototype-based Federated Learning, page 3)** of the main text **(see Equations 2 and 3)**. To improve readability, we will introduce a more detailed overview in the Introduction.
>
> > W2: GW Details
>
> Thank you for your question. We are happy to clarify this implementation detail:
>
> To ensure stability, efficiency, and the validity of the underlying mathematical solution, we directly utilized the industry-standard and widely adopted Python Optimal Transport (POT) open-source library. Specifically, we employed the `ot.gromov.gromov_wasserstein` interface to calculate the GW distance and perform topological alignment; consequently, we did not provide an extensive description in the main text. Regarding the Sinkhorn iteration count and the entropic regularization coefficient you inquired about, we strictly adhered to the official default parameter settings of this interface.
>
> 1. **Entropic Regularization Coefficient (ϵ):** In all experiments presented in this paper, we set `ϵ=0.1` by default. This value is a common empirical standard in GW literature.
> 2. **Sinkhorn Iterations:** We utilized the default stopping criterion of the POT library, defined by a relative tolerance of `tol=1e-9` and a maximum iteration limit of `max_iter=10000`. Our experimental logs indicate that the algorithm typically converges within `[50–300]` iterations (well below the upper limit), never reaching the maximum iteration count.
>
> To ensure the reproducibility of our research, we will provide a detailed explanation of these settings in the appendix.
>
> > W3: Method Details
>
> We appreciate the reviewer's constructive suggestion. Indeed, explicitly stating method names in the Related Work section helps highlight our contributions during the experimental comparison. We will emphasize the names associated with our baseline methods in Section 2.2. For example:
>
> - **FedProto** (Tan et al., 2022) pioneered the idea of transmitting class feature prototypes instead of model parameters.
> - **FedTGP** (Zhang et al., 2024) introduced trainable global prototypes and contrastive learning to handle data heterogeneity.
> - **FedPall** (Zhang et al., 2025c) utilizes adversarial and collaborative learning to mitigate feature shift issues.
>
> However, these methods exhibit the following limitations when facing model heterogeneity and modality imbalance:
>
> 1. **Knowledge Pollution from Incompatible Feature Spaces:** FedProto, FedTGP, and FedPall assume feature prototypes exist in a shared Euclidean space. However, heterogeneous client networks produce semantically distinct latent spaces. Directly averaging or calculating distances between these incompatible prototypes forces aggregation, disrupting decision boundaries and causing "knowledge pollution."
> 2. **Absence of Cross-Modal Bridges:** These baselines are designed for unimodal scenarios and lack mechanisms to align different modalities. Consequently, clients with rich modalities cannot transfer complementary knowledge to those with scarce modalities.
>
> In contrast, our method employs aligned Logit prototypes as a semantic bridge and uses GW distance to align relative topological structures rather than absolute coordinates, fundamentally overcoming these limitations in complex multimodal scenarios.
>
> > W4: Entropy Aggregation
>
> We appreciate the reviewer's professional inquiry. It is worth noting that, regarding the issue of incomparable heterogeneous Logit magnitudes, we did not directly use raw absolute Logit values to calculate entropy in **Section 4.1.1**. Instead, we first processed them via the **Softmax function**, as shown in **Eq. (5) (the right of line 179)**.
>
> > W5: Minor Issue
>
> We appreciate your keen observation. We will meticulously correct Fig. 1 to address this issue. Regarding Fig. 2, the confusion likely stems from our unclear description. We intended to convey that the dataset comprises $M$ modalities (e.g., Image1, Image2, Language, Audio, Video), whereas Client $k$ may contain only three modalities $|\mathcal{M}\_k|=3$ (e.g., Image1, Audio, Video). We apologize for the misunderstanding caused and will carefully revise the figure to provide a clearer illustration.
>
> > W6: Minor Issue
>
> We sincerely appreciate the reviewer's suggestions for clarification. We will follow the reviewer's advice to ensure consistent professional terminology and clear definitions throughout the manuscript.

---

> > ### Author Rebuttal · Reviewer_WGff · 2026-04-02
> >
> > I have raised my score from negative to positive.

---

> > > ### Author Response · Authors · 2026-04-03
> > >
> > > We sincerely thank you for your professional review and constructive feedback. We are truly grateful for the time and effort you have devoted to the review process.

---

### Official Review · Reviewer_r7Z3 · 2026-03-12

**Soundness:** 3
**Presentation:** 4
**Significance:** 4
**Originality:** 4
**Overall Recommendation:** 5
**Confidence:** 4

**Summary:**

This paper proposes a novel framework called "Prototype-Guided Bilateral Alignment Multimodal Federated Learning" (MFedPBA), which aims to address two core challenges in real-world multimodal federated learning (MFL) scenarios. This method promotes robust cross-client knowledge transfer by jointly processing heterogeneous feature spaces and aggregation decisions, while rigorously protecting the architectural heterogeneity of local models.

**Compliance With Llm Reviewing Policy:**

Affirmed.

**Final Justification:**

Thank you for the author's response. I agree with the author's analysis of privacy issues, so I have increased my score by 5.

**Key Questions For Authors:**

1. The paper mentions the difficulty of reconstructing the original input from the prototype. Can you provide corresponding analysis to support this claim?
2. Regarding communication costs, does the paper conduct a theoretical quantitative comparison to more rigorously demonstrate the advantages of prototype exchange?
3. For multimodal clients, when utilizing the server-side global prototype, besides simple feature summation, are there more advanced adaptive fusion methods?

If the concerns are adequately addressed during the rebuttal phase, I would be open to increasing my rating.

**Limitations:**

Yes

**Strengths And Weaknesses:**

Strengths:
1. The authors' research paper is significant, addressing crucial practical issues in current MFL research, namely model heterogeneity and modal imbalance, transcending the traditional homogeneity assumption and making the research more relevant to current realities.
2. The proposed bilateral alignment mechanism is ingeniously conceived. Feature-level alignment utilizes GW distance to handle the incompatibility of heterogeneous spaces, while decision-level alignment leverages the natural semantic sharing properties of logistic prototypes. Together, they form a complete and complementary knowledge fusion framework.
3. Extensive experimental results: Extensive experiments were conducted on four multimodal benchmark datasets and various data partitioning settings (M2, M1+, M1), demonstrating the superiority of the method under various modal imbalance conditions.

Weaknesses:
1. In the privacy analysis in the appendix, the paper points out that MFedPBA enhances privacy by exchanging only prototypes (rather than the original data or gradients) and demonstrates the difficulty of reconstructing the original input from the prototypes. Is there a corresponding mathematical explanation for this?
2. Compared to traditional federated learning, prototype communication does reduce communication costs, but the paper lacks a theoretical proof by comparing it with other methods.
3. The paper mentions that during local training, for clients with multimodalities, the strategy is to simply sum the feature vectors of different modalities before feeding them into the classifier. Although the paper mentions that preliminary experiments show that "sum" is superior to "concat", this is a relatively rudimentary fusion method. Could the paper discuss how to utilize the server-side aligned global prototypes to guide the client in more advanced, adaptive multimodal fusion (such as attention mechanisms)?

---

> ### Author Rebuttal · Authors · 2026-03-28
>
> Thank you to the reviewer for their questions regarding privacy efficiency, communication volume, and integration methods. We will now answer these questions one by one.
>
> > W1Q1: Regarding Privacy Analysis
>
> We sincerely appreciate your interest in the privacy-preserving mechanisms discussed in our appendix. Prototype learning is widely recognized in the federated learning community for offering enhanced privacy guarantees [1-2]. We explain this mechanism through mathematical and logical facts across two distinct layers:
>
>  **1) Mathematical Irreversibility via Aggregation:**
> This represents the most intuitive and core mathematical barrier. In the MFedPBA framework, the feature and logit prototypes uploaded to the server are not outputs of single samples, but rather the standard weighted averages of all local samples within the same category：
>
> $$E\_{k,c}^{m} = \frac{1}{|\mathcal{D}\_{k,c}^m|} \sum\_{(\boldsymbol{x},y)\in \mathcal{D}\_{k,c}^m} f\_k^m({\boldsymbol{x}}).$$
>
> Mathematically, this constitutes a many-to-one mapping. If an attacker attempts to infer individual sample features, they face a severely underdetermined system of equations. With only one mean vector as the known quantity and $N$ independent feature vectors as unknowns, the system possesses infinite solutions whenever $N>1$ . Consequently, it is mathematically impossible for an attacker to isolate or extract specific individual data features from a mean; unique individual information is thoroughly "submerged" within the statistical distribution of the group during the averaging process.
>
> **2) "Black Box" Defense via Model Heterogeneity:**
> In traditional federated learning, clients upload gradients or weights, and the server possesses the global model architecture. This allows attackers to perform "model inversion attacks" to reconstruct original images via backpropagation. However, in MFedPBA, such attacks are mathematically infeasible due to model heterogeneity. Specifically, the architecture of the client-side feature extractor remains completely confidential to the server and external entities. An attacker must know the specific network structure and all weight parameters of $f\_k^m(\cdot)$ to compute $\frac{\partial f}{\partial x}$ , which is impossible in our setting.
>
> [1] Fedtgp: Trainable global prototypes with adaptive-margin-enhanced contrastive learning for data and model heterogeneity in federated learning, AAAI 2024.
>
> [2] Fedpall: prototype-based adversarial and collaborative learning for federated learning with feature drift. CVPR 2025.
>
> > W2Q2: Discussion on Communication
>
> We appreciate the reviewer's comment. We have provided the corresponding theoretical complexity analysis on **page 7 (Communication Efficiency section)**. However, due to space constraints, we were unable to detail the communication volumes of other methods. It is worth emphasizing that prototype-based methods incur significantly lower upload costs compared to methods that directly upload model parameters.
>
> For instance, the parameter-based method Harmony has a per-round communication complexity of $\sum\_{k}(M\_k \theta\_k)$ . Compared to millions of model parameters $\theta\_k$, the dimensionality of prototypes is smaller by several orders of magnitude. Among prototype-based methods, FedProto and FedTGP involve only prototype communication, denoted as $K2CM\_kd\_D$ , whereas FedPall requires uploading the classifier, denoted as $K(2CM\_kd_D+h\_k)$ .
>
> Our method, MFedPBA, transmits feature prototypes with the addition of only one set of Logit prototypes. Notably, since Logit prototypes represent a cross-modal shared naturally aligned semantic space, they do not need to be multiplied by the modality count $M\_k$ . The theoretical two-way communication complexity is $K2C(M\_k d\_D + d\_C)$ . This additional overhead accounts for a negligible proportion of the total bandwidth usage.
>
> >W3Q3: Discussion on Integration Forms
>
> We thank the reviewer for the insightful suggestion. We agree that advanced fusion mechanisms like attention could further improve performance. However, we wish to clarify that the core contribution of MFedPBA lies in addressing cross-node collaboration challenges, specifically model heterogeneity and modality imbalance, rather than innovating local fusion architectures.
>
> We intentionally employ the most fundamental fusion strategy on local clients primarily for experimental considerations regarding controlled variables and fair comparison. To rigorously demonstrate that the performance improvements stem purely from our proposed server-side prototype alignment and client-side knowledge distillation mechanisms, we need to isolate the gains attributable to the complexity of local network architectures. Introducing complex **attention** or **gating** fusion mechanisms locally could obscure the specific contributions of the federated knowledge transfer framework itself.

---

> > ### Author Rebuttal · Reviewer_r7Z3 · 2026-04-02
> >
> > Thank you for the author's response. I agree with the author's analysis of privacy issues, so I have increased my score by 5.

---

> > > ### Author Response · Authors · 2026-04-03
> > >
> > > Thank you again for your valuable suggestions and insightful comments. We are sincerely grateful that you took the time out of your busy schedule to review our paper.

---

### Decision · Program_Chairs · 2026-04-30

**Decision:**

Accept (spotlight)

**Comment:**

This paper proposes a novel framework called "Prototype-Guided Bilateral Alignment Multimodal Federated Learning" (MFedPBA), addressing two core challenges in real-world multimodal federated learning (MFL) scenarios. The problem is significant and the method is novel.